# Zooming from Context to Cue: Hierarchical Preference Optimization for Multi-Image MLLMs

**Xudong Li**[1][*]    **Mengdan Zhang**[2][*]  **Peixian Chen**[2]   **Xiawu Zheng**[1]   **Yan Zhang**[1][†]
**Jingyuan Zheng**[1]    **Yunhang Shen**[2]    **Ke Li**[2]    **Chaoyou Fu**[3]    **Xing Sun**[2]    **Rongrong Ji**[1]

[1] Key Laboratory of Multimedia Trusted Perception and Efficient Computing,
Ministry of Education of China, Xiamen University, 361005, P.R. China
[2] Tencent Youtu Lab    [3] Nanjing University
{lxd761050753, zhangmengdanrz}@gmail.com, {zhengxiawu, rrji}@xmu.edu.cn

## Abstract

Multi-modal Large Language Models (MLLMs) excel at single-image tasks but struggle with multi-image understanding due to cross-modal misalignment, leading to hallucinations (context omission, conflation, and misinterpretation). Existing methods using Direct Preference Optimization (DPO) constrain optimization to a solitary image reference within the input sequence, neglecting holistic context modeling. To address this, we propose **C**ontext-to-**C**ue **D**irect **P**reference **O**ptimization (**CcDPO**), a multi-level preference optimization framework that enhances per-image perception in multi-image settings by zooming into visual clues— from sequential context to local details. Our approach features two sequentially dependent components: (i) *Context-Level Optimization:* By introducing low-cost sequence preference pairs, we optimize the model to distinguish between complete and disrupted multi-image contexts, thereby correcting cognitive biases in MLLMs' multi-image understanding. (ii) *Needle-Level Optimization:* By integrating region-specific visual prompts with multimodal preference supervision, we direct the model's attention to critical visual details, effectively suppressing perceptual biases toward fine-grained visual information. To support scalable optimization, we also construct **MultiScope-42k**, an automatically generated multi-image dataset with hierarchical preference pairs. Experiments show that CcDPO significantly reduces hallucinations and yields consistent performance gains across general single- and multi-image tasks. Codes are available at https://github.com/LXDxmu/CcDPO.

## 1 Introduction

Simultaneously understanding multiple images remains a fundamental yet underexplored challenge for Multi-modal Large Language Models (MLLMs) [1, 2, 3, 4]. Despite MLLMs excelling in single-image tasks like visual question answering (VQA) [5, 6, 7], code generation [8, 9], and storytelling [10, 11], and open-source models such as LLaVA [12], BLIP-2 [13], and InternVL [14] showing competitive results on benchmarks including VQAv2 [15], OKVQA [16], and MMMU [17], their capabilities in multi-image contexts are notably constrained. These models frequently struggle with tasks demanding cross-image comparison, spatial reasoning, or temporal alignment [18], often resulting in hallucinations like context omission, conflation, and misinterpretation of local details. These deficiencies ultimately compromise model reliability. The root cause lies in the weak cross-modal alignment within MLLMs, which frequently fails to integrate visual and textual information

---

[*]Equal Contribution.
[†]Corresponding Author.

39th Conference on Neural Information Processing Systems (NeurIPS 2025).

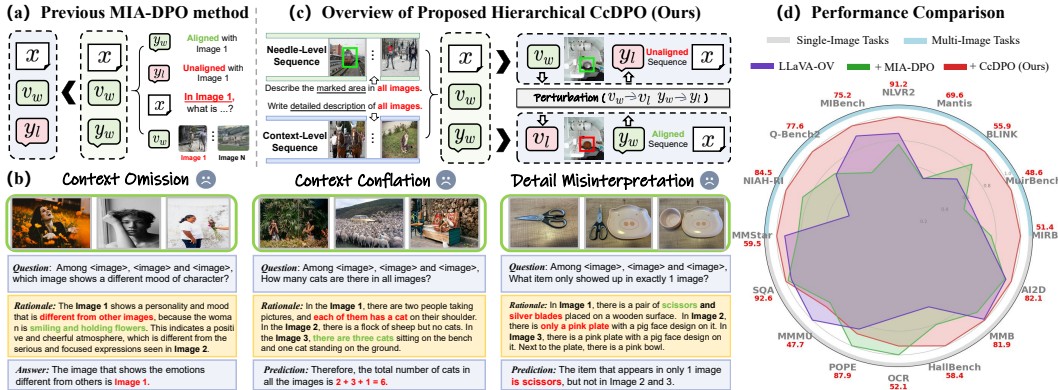

Figure 1: (a) Prior multi-image DPO (e.g., MIA-DPO) is constrained by its reliance on predefined image references and text-only preferences, limiting holistic context modeling. (b) These limitations commonly lead to failures such as Context Omission (ignoring relevant images), Context Conflation (misattributing content across images), and Detail Misinterpretation (misrepresenting fine-grained visual cues). (c) CcDPO addresses these issues by hierarchically enhancing MLLMs' visual perception, from overall multi-image contexts to specific fine-grained details. (d) Benchmark comparisons demonstrate CcDPO's improved reasoning capabilities on both multi-image and single-image tasks.

coherently and comprehensively. This limitation becomes particularly pronounced in multi-image settings, where accurate reasoning requires both: (i) Precise interpretation of intra-image regional details, and (ii) establishing meaningful inter-image connections through contextual integration.

To overcome these limitations, instruction tuning with multi-image supervision has been adopted in recent models such as Flamingo [19], IDEFICS [20], and Emu2 [21]. However, these approaches rely on large-scale annotated data, which is costly to construct due to the complexity of modeling inter-image relationships. As a lightweight alternative, *Direct Preference Optimization* (DPO) [22] has emerged as a promising training paradigm, aligning model outputs with human preferences through pairwise supervision without requiring large-scale labeled data, significantly reducing reliance on costly annotations. Recent work has extended DPO to multimodal tasks [23, 24, 25, 26], with MIA-DPO [27] specifically pioneering its application to multi-image scenarios. As shown in Fig. 1(a), MIA-DPO conditions responses on a specific image through explicit query references (e.g., "In Image 1, what is...?"), helping the model associate questions with the correct visual input in context.

While this anchoring strategy mitigates referential ambiguity in MLLMs, the absence of explicit contextual modeling in its architectural design fundamentally limits comprehensive cross-image integration. This constraint is further exacerbated by the high vulnerability of inter-image attention mechanisms to multi-image interference [28, 29, 30]. Consequently, without explicit image references, such approaches often demonstrate limited capacity in autonomously capturing sequential visual context and fine-grained details, giving rise to multiple forms of multi-image hallucinations: **Context Omission**: The model selectively ignores subsets of input images, generating responses based on incomplete sequences (e.g., ignoring Image 3; Fig. 1(b), left). **Context Conflation**: The model erroneously attributes visual elements across images (e.g., describing a cat from Image 3 as appearing in Image 1; Fig. 1(b), middle). **Detail Misinterpretation**: Critical visual details in a certain image are either missed or misinterpreted (e.g., without explicit image-specific instructions, the model fails to recognize the scissors in Image 2 and erroneously detects silver blades in Image 1; Fig. 1(b), right).

To address these challenges, we propose **C**ontext-to-**C**ue **D**irect **P**reference **O**ptimization (**CcDPO**), a *two-level preference optimization* framework that enhances MLLMs' capability to accurately perceive visual information across hierarchical levels—from sequential multi-image contexts to individual fine-grained details (as shown in Fig. 1(c)). Specifically, it consists of two levels of alignment:

**(1) Context-Level Optimization:** We formulate structured multi-image captioning as a proxy task to enforce global contextual alignment. By explicitly decomposing model responses into per-image descriptions (e.g., "For Image 1: <caption 1>", "For Image 2: <caption 2>"), we compel the model to accurately attend to each image within the sequence. This structured formulation ensures contextual completeness while mitigating inter-image interference. To further address **Context Omission** and **Context Conflation**, we introduce two perturbation techniques—*sequence truncation* and *content*

*swapping*—into the captioning preference optimization process. By training the model to distinguish coherent contexts from disrupted ones, we promote holistic reasoning across the entire input sequence.

**(2) Needle-Level Optimization:** To address **Detail Misinterpretation**, we propose a fine-grained preference learning strategy that sharpens the model's sensitivity to critical visual cues. Our approach integrates region-focused visual prompts into the preference data and employs DPO training to bias the model toward descriptions aligned with highlighted regions. This enhances the model's ability to detect, attend to, and describe salient visual elements across multiple images. Furthermore, inspired by [31], we incorporate vision contrastive preference supervision by constructing image pairs with varying alignment to reference descriptions. This encourages the model to refine its preference judgments on fine-grained visual cues within each image under contextual settings.

To support these two-level optimization objectives, we introduce **MultiScope-42k**, a scalable multi-image preference dataset. The dataset comprises high-quality *chosen* responses—synthesized by splicing together accurate image- and region-level descriptions—alongside *rejected* responses generated through targeted perturbations at both contextual and local detail levels. This pipeline converts abundant single-image data into complex multi-image training signals, providing a scalable solution to data scarcity. Our main contributions are summarized as follows:

- We pioneer the investigation of cognitive bias in multi-image comprehension for MLLMs, categorizing three prevalent hallucination types. To address these challenges, we propose **C**ontext-to-**C**ue **D**irect **P**reference **O**ptimization (**CcDPO**), an innovative two-level preference optimization framework that enhances per-image perception in multi-image settings by analyzing visual clues—from sequential context to local details.

- We design a low-cost Context-Level Optimization mechanism, incorporating structured multi-image captioning preferences and targeted perturbation techniques to ensure MLLMs' comprehensive and consistent global context understanding. Complementarily, we develop a Needle-Level Optimization mechanism that enhances fine-grained visual acuity through the integration of region-focused visual prompts and vision contrastive preference signals.

- We construct **MultiScope-42k**, a large-scale, high-quality dataset for two-level multi-image preference learning. The fully automatic generation pipeline is cost-effective and scalable across diverse data sources. After direct preference optimization on this dataset, our method significantly reduces hallucinations and achieves superior performance on multi-image tasks.

## 2 Related Work

**Multi-modal Large Language Models.** Recent advances in MLLMs [12, 32, 33, 34] have combined powerful large language models (LLMs) with visual encoders via lightweight connectors, achieving impressive performance across dialogue [35], visual question answering (VQA) [5], and image captioning tasks [36]. These models are typically trained on image-text pairs with instruction tuning, yielding strong single-image understanding. However, they remain prone to hallucinations [37, 38, 39], especially in multi-image scenarios where accurate reasoning requires modeling not only individual images but also their cross-image relationships. Recent studies aim to advance multi-image understanding by incorporating image-text interleaved data [40, 41] during model training. This approach helps develop capabilities such as image comparison [42, 43], cross-image association [44, 45], and temporal reasoning [46, 47]. Nevertheless, instruction tuning with such data remains costly due to the need for complex, fine-grained annotations—an issue exacerbated in multi-image settings.

**Direct Preference Optimization.** Reinforcement Learning from Human Feedback (RLHF) [48, 39, 3] aligns LLMs with human preferences by training a reward model to maximize the gap between chosen and rejected responses. As a more efficient alternative, Direct Preference Optimization (DPO) [22] bypasses reward modeling and directly optimizes on preference pairs. Recent work investigates DPO's generalization and stability across tasks [26, 49, 50, 51], with multimodal extensions [31, 52, 53, 26] to reduce hallucinations and enhance vision-language grounding in single-image settings. However, current language-based DPO methods often neglect visual details. To address this, vision contrastive DPO approaches either disrupt images [23, 52] or highlight key visual tokens [54, 55], enhancing preference learning but focusing mainly on single-image tasks. MIA-DPO [27] pioneered DPO to multi-image settings by anchoring prompts to specific images, achieving promising results on relevant benchmarks. However, its reliance on predefined references limits holistic context modeling and autonomous cross-image reasoning. In contrast, we propose CcDPO, which explicitly models global context and fine-grained visual cues to enhance multi-image reasoning.

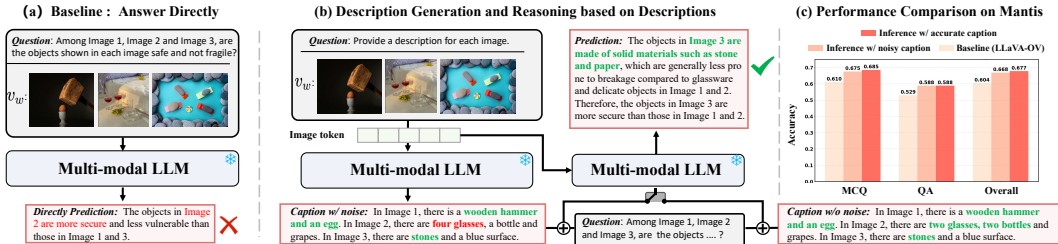

Figure 2: (a) Baseline: Direct inference without context as a condition. (b) Two-stage approach: Generating image captions, then reasoning over them. (c) Performance: Accurate caption understanding as context substantially improves VQA accuracy, with noisy captions also proving beneficial. This highlights deficient intrinsic captioning in MLLMs as a key bottleneck, motivating its enhancement.

**Visual Prompting for MLLMs.** Visual prompting has been widely used in vision models [56, 57, 58]. Manually annotated points, boxes, or masks—often encoded by a separate prompt encoder—can guide the model to adjust segmentation granularity or select specific instances. More recently, MLLMs have shown the ability to interpret visual prompts directly embedded in the image without additional prompt encoders [59, 60]. Unlike prior work, our method actively integrates visual prompts into preference data and uses DPO training to encourage the model to prefer descriptions aligned with prompted regions, which enhances the model's sensitivity to visually grounded information.

## 3 Exploring Cognitive Bias in Multi-Image Comprehension of MLLMs

In this section, we investigate the cognitive biases of MLLMs in multi-image comprehension, identifying that deficient internal multi-image captioning ability is a core bottleneck for complex reasoning. Then, we introduce a systematic evaluation methodology to diagnose and quantify three fundamental multi-image hallucination types. Our results confirm that even state-of-the-art models exhibit severe performance degradation and elevated hallucination rates as input images increase.

**Limited Contextual Awareness.** We identify a fundamental limitation in current MLLMs: their impaired ability to perceive and integrate partial context information for coherent multi-image understanding severely degrades performance. To investigate this limitation, we systematically examine how context quality affects multi-image understanding through controlled experiments with LLaVA-OV-7B [32]. As illustrated in Fig. 2, we evaluate the model's reasoning capability under three distinct context conditions: (1) Accurate Context: The model received the images along with accurate, descriptive captions. (2) Noisy Context: It received the images along with flawed, machine-generated captions. (3) Baseline (No Explicit Context): It received only the images, forcing it to rely entirely on its own internal understanding. As shown in Fig. 2 (c), while accurate captions helped slightly more than noisy ones, the most dramatic finding was a massive **7.37-point accuracy drop** when the model performed inference directly from the images, without any context augmentation. This gap reveals the model's heavy reliance on external caption aids, confirming that its intrinsic captioning capability is a core bottleneck for multi-image reasoning and motivating our direct efforts to enhance this skill.

**Multi-image Captioning Re-evaluation.** We design a caption generation task as a proxy to systematically evaluate MLLMs' multi-image understanding, identifying three fundamental hallucination types—*Context Omission*, *Context Conflation*, and *Detail Misinterpretation*—that critically degrade model performance. To enable this evaluation, we construct **Context-AMBER-1K** by systematically concatenating images from the single-image AMBER dataset [37] into sequences of two types: short-context sequences with 4 images and long-context sequences with 8 images. Each input is paired with the prompt: ``Please sequentially describe each of the images shown above. Use the following format: For Image *:⟨description⟩." The expected output format is: ``For Image 1:⟨caption 1⟩,For Image 2:⟨caption 2⟩,..., For Image N:⟨caption N⟩.", ensuring comprehensive coverage of all input images. This controlled format allows precise identification of hallucination behaviors during multi-image comprehension.

We assess caption quality through four complementary metrics: (a) CHAIR [36] measures object hallucination severity, (b) Response-level Hallucination Rate (Hal) [37] quantifies incorrect descriptions, (c) Cognition-based Hallucination (Cog) [37] detects reasoning errors, and (d) Sequence Coverage Rate (SCover) evaluates caption completeness across image sequences. Notably, for each input sequence, we compute hallucination scores (a), (b), and (c) for each image independently, then average these to get the sequence-level hallucination scores.

Table 1: **Hallucination and preference alignment results.** We report metrics on our constructed multi-image AMBER benchmark. Lower scores indicate better performance for CHAIR, Hal, and Cog, while higher is better for SCover. CcDPO achieves consistent improvements under both 4-image and 8-image settings, effectively reducing hallucinations in contextual multi-image understanding.

| Models | Parameter | Context-AMBER (4 Images) | | | | Context-AMBER (8 Images) | | | |
|---|---|---|---|---|---|---|---|---|---|
| | | CHAIR↓ | SCover↑ | Hal↓ | Cog↓ | CHAIR↓ | SCover↑ | Hal↓ | Cog↓ |
| LLaVA-OV | 7B | 10.2 | 74.0% | 31.8 | 2.6 | 50.6 | 10.3% | 69.1 | 6.5 |
| + MIA-DPO [27] | 7B | 8.9 | 83.9% | 29.8 | 2.1 | 28.2 | 36.7% | 45.0 | 3.8 |
| + CcDPO (Ours) | 7B | **3.7** | **100.0%** | **15.3** | **1.2** | **15.3** | **83.3%** | **27.5** | **2.1** |
| Δ | - | **+6.5** | **+26.0** | **+16.5** | **+1.4** | **+35.3** | **+73.0** | **+41.6** | **+4.4** |

**Results.** As shown in Table 1, even strong models like LLaVA-OV-7B exhibit significant hallucination rate increases in multi-image scenarios. For example, when input images grow from 4 to 8, *Detail Misinterpretation hallucinations* become severe—the CHAIR score jumps from **10.2 to 50.6**, indicating a sharp decline in grounding accuracy. The sharp drop in the SCover score from **74.0% to 10.3%** also reveals *context omission issues* within multi-image settings. Analysis of failure cases (Fig. 1) reveals that *Context Conflation* commonly occurs, critically degrading model performance.

# 4   CcDPO: Context-to-Cue Direct Preference Optimization

As discussed in Sec. 3, three fundamental hallucinations (context omission, context conflation, and detail misinterpretation) manifest as performance degradations in MLLMs' multi-image understanding. To address these challenges, we propose **C**ontext-to-**C**ue **D**irect **P**reference **O**ptimization (**CcDPO**), a hierarchical preference alignment framework that refines MLLMs at two levels (as shown in Fig. 3):

• **Context-Level Optimization:** By contrasting complete and disrupted multi-image captions using language-based preference optimization, we enhance MLLMs' contextual understanding by ensuring comprehensive integration of all relevant visual information across image sequences.

• **Needle-Level Optimization:** A hybrid preference optimization method integrates two complementary objectives: (1) Contrasts captions that either align with or mismatch visually prompted regions using language-based preference optimization, and (2) Discriminates between images semantically matching or contradicting given captions using vision contrastive preference optimization. This dual approach trains the model to make preference judgments grounded in fine-grained visual details.

To support such hierarchical DPO, we construct **MultiScope-42k**, a large-scale preference dataset with automatically generated positive and perturbed response pairs at both levels. See Appendix A.

## 4.1   Context-Level DPO with Language-Based Preference Optimization

We propose a low-cost DPO mechanism for MLLMs that leverages multi-image captioning as a proxy task to enforce coherent multi-image understanding, thereby addressing context hallucinations (omission, conflation). Specifically, we reformulate the response generation task as a structured, per-image captioning problem. Each image in a sequence is described in an explicit format, generating the preferred response $y_w$ as a coherent sequence of captions that reflect the content of each image:

$$y_w = \{\texttt{For Image 1:} \langle \texttt{caption 1} \rangle, \texttt{For Image 2:} \dots, \texttt{For Image N:} \langle \texttt{caption N} \rangle.\}$$

This encourages the model to practice selective attention and attribution-aware generation. During training, we construct preference pairs where the **positive sample** is a coherent, full-sequence description $y_w$, and the **negative sample** $y_l$ is obtained from following two perturbation strategies:

• **Sequence Truncation**: simulates context omission by either removing captions entirely from one or more images (complete content omission) or replacing detailed captions with shorter versions (partial content omission), generating rejected responses $y_l^{\text{trunc}}$ and $y_l^{\text{short}}$. Complete omission disrupts sequence continuity, while partial omission results in sparse or incomplete sequence information:

$$y_l^{\text{trunc}} = \{\texttt{For Image 1:} \langle \texttt{caption 1} \rangle, \texttt{For Image 3:} \langle \texttt{caption 3} \rangle, \dots\}$$
$$y_l^{\text{short}} = \{\texttt{For Image 1:} \langle \text{ short caption 1 } \rangle, \texttt{For Image 2:} \langle \text{ short caption 2 } \rangle, \dots\}$$

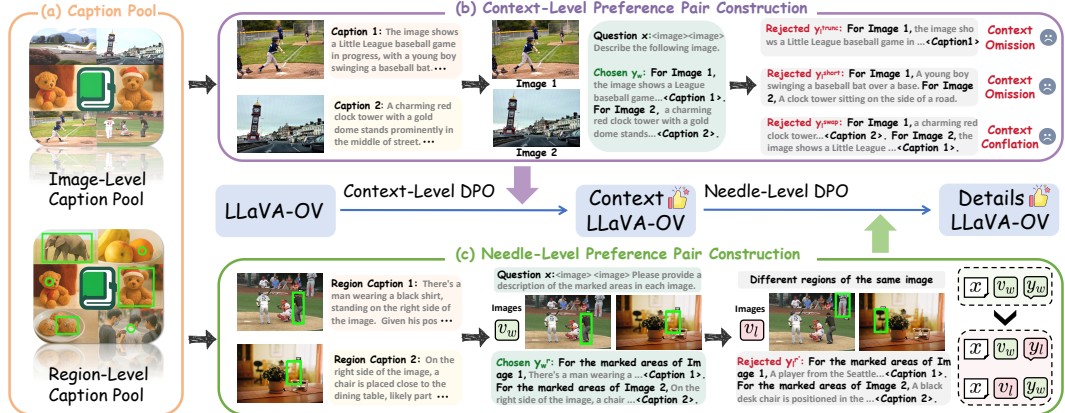

Figure 3: **Overview of CcDPO.** (a) Caption pools are built from LLaVA-23K [61], MDVP [63], and MVC [49] for image- and region-level supervision. (b) Context-Level DPO aligns model outputs with complete, coherent image sequences and penalizes omissions, conflation, and misalignments. (c) Needle-Level DPO incorporates visual prompts to enhance local detail understanding. chosen responses describe marked regions accurately, while rejected are drawn from mismatched regions. Both language-based and vision-contrastive preferences are used to sharpen fine-grained perception.

- **Content Swapping**: simulates context conflation by swapping the captions between different images within the sequence, thereby generating the rejected response $y_l$. e.g.,

$$y_l^{\text{swap}} = \{\texttt{For Image 1:} \langle \texttt{caption 2} \rangle, \texttt{For Image 2:} \langle \texttt{caption 1} \rangle, \ldots\}$$

Given an instruction $x$ and image sequence $v_w$, we define the context-level DPO objective as:

$$\mathcal{L}_{\text{DPO}_t} = -\log \sigma \left( \beta \log \frac{\pi_\theta(y_w \mid v_w, x)}{\pi_{\text{ref}}(y_w \mid v_w, x)} - \beta \log \frac{\pi_\theta(y_l \mid v_w, x)}{\pi_{\text{ref}}(y_l \mid v_w, x)} \right), \quad y_l \in \{y_l^{\text{trunc}}, y_l^{\text{short}}, y_l^{\text{swap}}\}$$

(1)

where $\pi_\theta$ is the target model and $\pi_{\text{ref}}$ is a frozen reference model. This objective reinforces the preference $(x, v_w, y_w) \succ (x, v_w, y_l)$ for globally coherent responses over disrupted ones. We use LLaVA-23K [61] and COCO [62] as our detailed and brief context caption pool, respectively. The visualization cases for three different types of context preferences are presented in Figs. 6, 7, and 8.

## 4.2 Needle-Level DPO with Hybrid Visual-Language Optimization

Even when global context is preserved, MLLMs often fail to identify or attend to salient visual elements (e.g., missing objects, actions, or attributes). This leads to **detail misinterpretation**, which global response-based supervision alone cannot resolve. To address this, we introduce **needle-level optimization**, a fine-grained hybrid preference mechanism employing visual prompts and image-level perturbations to sharpen the model's focus on often-overlooked local visual cues.

**Language-based Preference Optimization (TDPO).** This stage leverages region-specific visual prompts (bbox and point) to guide the model's preference judgments. The visual prompt construction involves three key steps: (1) collecting source data with point or bounding box annotations, where points are defined as [x, y] coordinates and bounding boxes as [x1, y1, x2, y2] corner coordinates; (2) converting these annotations into visual prompts (e.g., red points or green rectangles) overlaid onto original images; (3) enhancing region salience by adding textual labels (e.g., "REF") adjacent to visual markers, as illustrated in Fig. 3. These visual prompts explicitly define the target regions for description. To train the model to interpret these specified areas accurately, our DPO method learns from preference pairs that penalize misinterpretations. These pairs are constructed as follows:

**1) Chosen Responses ($y_w^r$):** We integrate region-level visual prompts (e.g., bounding boxes, key-points) into images $v$ to highlight a target region $r$, yielding $v_r$. The model is trained to prefer the accurate description $y_w^r$ of this specific region, directing its attention to critical visual elements:

$$y_w^r = \{\texttt{For the marked area of Image 1:} \langle \texttt{caption } r_1 \rangle, \texttt{For the marked area of Image 2:} \langle \texttt{caption } r_2 \rangle, \ldots, \texttt{For the marked area of Image N:} \langle \texttt{caption } r_N \rangle.\}$$

**2) Rejected Responses ($y_l^{r'}$):** Descriptions of regions $r'$ that are non-overlapping with $r$ within the same image serve as rejections. By learning to identify and reject such region-specific inaccuracies, the model's perception of fine-grained image details is more precise. $y_l^{r'}$ is formally defined as:

$$y_l^{r'} = \{\texttt{For the marked area of Image 1: } \langle\texttt{caption } r_1'\rangle, \texttt{For the marked area of Image 2: } \langle\texttt{caption } r_2'\rangle,..., \texttt{For the marked area of Image N: } \langle\texttt{caption } r_N'\rangle.\}$$

Given an instruction $x$ and the image sequence $v_r$, these pairs $(x, v_r, y_w^r) \succ (x, v_r, y_l^{r'})$ inform the language-based DPO objective (Eq. 1). We utilize MDVP [63] for the region-level caption pool.

**Vision Contrastive Preference Optimization (VDPO).** Inspired by [49, 31], this stage further hones the model's visual discrimination. It trains the model by contrasting a single description $y_w$ against two image inputs: $v_w$, which correctly aligns with $y_w$ (often focusing on a specific visual region), and $v_l$, which is misaligned or visually contradicts $y_w$. The objective combines two components:

**1) Focusing on Relevant Visuals ($\mathcal{L}_{\textbf{Focus}}$):** This rewards prioritizing details in the correctly aligned image $v_w$ when generating $y_w$, countering MLLMs' tendency to neglect visual content.

$$\mathcal{L}_{\text{Focus}}(v_w, y_w) = -\log\sigma\left(\beta_1\log\frac{\pi_\theta(y_w \mid v_w, x)}{\pi_{\text{ref}}(y_w \mid v_w, x)} - \beta_1\log\frac{\pi_\theta(y_w \mid x)}{\pi_{\text{ref}}(y_w \mid x)}\right), \quad (2)$$

**2) Rejecting Contradictory Visuals ($\mathcal{L}_{\textbf{Reject}}$):** This penalizes assigning high probability to $y_w$ when conditioned on a contradictory image $v_l$.

$$\mathcal{L}_{\text{Reject}}(v_l, y_w) = -\log\sigma\left(\beta_2\log\frac{\pi_\theta(y_w \mid x)}{\pi_{\text{ref}}(y_w \mid x)} - \beta_2\log\frac{\pi_\theta(y_w \mid v_l, x)}{\pi_{\text{ref}}(y_w \mid v_l, x)}\right), \quad (3)$$

The combined vision contrastive DPO loss is $\mathcal{L}_{\text{DPO}_v}(v_w, y_w, v_l) = \mathcal{L}_{\text{Focus}}(v_w, y_w) + \mathcal{L}_{\text{Reject}}(v_l, y_w)$. This objective sharpens the model's ability to distinguish fine-grained visual cues by rewarding focus on relevant details and penalizing attention to misleading content. We use the MVC [49] dataset as a region-level visual counterfactual caption pool. Visualization cases are shown in Figs. 9, 10, and 11.

## 5 Experiments

### 5.1 Experimental Settings and Evaluation Benchmarks

**Baselines.** We apply CcDPO to two different 7B-size MLLMs: Qwen2-VL [33] and LLaVA-OV [32]. Due to differences in base models, preference data, and alignment strategies, direct comparisons with other LLMs are not possible. However, we provide the results for reference: LLaVA-1.5 [64], InternVL2-8B [65], Mantis-Idefics [44], mPLUG-Owl3 [66], Idefics2-8B [20], and Emu2-Chat [21].

**Implementation Details.** Our model undergoes a three-stage sequential training process to better understand multi-image preferences at both broad (context) and detailed (needle) levels. **Stage 1** focuses on context-level alignment, where we fine-tune Qwen2-VL-7B and LLaVA-OV-7B for one epoch with learning rates of $5 \times 10^{-6}$ and $5 \times 10^{-5}$, respectively, using Eq. 1. **Stage 2** applies needle-level language-based DPO using Eq. 1 to improve sensitivity to fine-grained visual cues with the same learning rate of $5 \times 10^{-5}$. We conduct Stage 1 and Stage 2 by using LoRA adaptation [67] with rank $r = 128$ for efficiency. **Stage 3** performs vision contrastive DPO via full-parameter tuning for one epoch with a learning rate of $1 \times 10^{-6}$ using Eq. 2, strengthening the model's ability to distinguish preferred visual content. Following the setup in [27], we set the temperature parameter $\beta = \beta_1 = \beta_2 = 0.1$ and the negative log-likelihood (NLL) loss coefficient $\gamma = 0.1$. All training is conducted on eight GPUs, each equipped with 90GB of memory.

**Evaluation Benchmarks.** We employ seven multi-image benchmarks—MUIRBench [42], MIRB [68], BLINK [69], Mantis-Eval [44], NLVR2 [70], Q-Bench2 [71], and MIBench [72]—to holistically evaluate multi-image reasoning across four key dimensions: co-reference alignment, fine-grained comparison, contextual reasoning, and temporal understanding. Complementing these, eight representative single-image benchmarks assess specific multimodal capabilities: (1) Academic/Scientific Reasoning: MMMU [73], MMStar [74], ScienceQA [75], (2) Diagram Understanding: AI2D [76], (3) Robustness against hallucinations: POPE [77], HallBench [38], (4) General Multimodal Abilities: MMBench [78], (5) Text Recognition: OCRBench [79]. This comprehensive evaluation suite demonstrates our method's strengths in both holistic understanding and fine-grained visual grounding across single-image and multi-image general tasks.

Table 2: **Main results on general multi-image benchmarks.** We compare our proposed method, CcDPO, with existing multi-image DPO approaches across seven multi-image benchmarks. Our method consistently enhances the performance of both LLaVA-OV and Qwen2-VL.

| Models | Parameter | MuirBench | MIRB | BLINK | Mantis | NLVR2 | MIBench | Q-Bench2 | Average |
|---|---|---|---|---|---|---|---|---|---|
| GPT-4o [80] | - | 62.3 | 53.0 | 60.1 | 62.7 | 88.8 | 71.8 | 74.5 | 67.6 |
| LLaVA-v1.5 [64] | 7B | 19.9 | 28.4 | 37.1 | 41.9 | 52.1 | 40.9 | 53.9 | 39.2 |
| Idefics2 [20] | 8B | 26.1 | 33.0 | 45.2 | 48.9 | 86.9 | 29.7 | 57.0 | 46.6 |
| Mantis-Idefics2 [44] | 8B | 44.5 | 41.8 | 49.1 | 57.1 | 89.7 | 44.3 | 75.3 | 57.4 |
| mPLUG-Owl3 [66] | 8B | 39.6 | - | 50.3 | 63.1 | 90.8 | 54.5 | - | 59.6 |
| Emu2-Chat [21] | 37B | 33.6 | 27.2 | 36.2 | 37.8 | 58.2 | 39.7 | 65.3 | 42.6 |
| InternVL2-8B [65] | 8B | 48.7 | 50.0 | 50.6 | 60.3 | 85.56 | 52.9 | - | 58.0 |
| LLaVA-OV [32] | 7B | 42.5 | 47.3 | 51.1 | 60.4 | 89.4 | 73.6 | 73.8 | 62.5 |
| + SFT | 7B | 45.4 | 48.9 | 53.4 | 64.9 | 89.0 | 71.9 | 75.7 | 64.1 |
| + MIA-DPO [27] | 7B | 41.4 | 48.0 | 53.7 | 60.3 | 88.2 | 67.8 | 74.0 | 61.9 |
| + CcDPO (Ours) | 7B | **48.6** | **51.4** | **55.9** | **69.6** | **91.2** | **75.2** | **77.6** | **67.1** |
| Δ | - | **+6.1** | **+4.1** | **+4.8** | **+9.2** | **+1.8** | **+1.6** | **+3.8** | **+4.6** |
| Qwen2-VL [33] | 7B | 40.5 | 59.5 | 53.4 | 65.9 | 84.8 | 68.9 | 74.5 | 63.9 |
| + SFT | 7B | 43.1 | 59.8 | 54.7 | 64.9 | 85.2 | 69.4 | 74.1 | 64.5 |
| + MIA-DPO [27] | 7B | 40.1 | **61.4** | 54.5 | 69.1 | 84.5 | 66.7 | 75.6 | 64.5 |
| + CcDPO (Ours) | 7B | **44.8** | 60.7 | **56.5** | 69.1 | **86.4** | 71.9 | **77.0** | **66.6** |
| Δ | - | **+4.3** | **+1.2** | **+3.1** | **+3.2** | **+1.6** | **+3.0** | **+2.5** | **+2.7** |

Table 3: **Main results on single-image benchmarks.** We compare our CcDPO with existing DPO-based approaches across seven single-image benchmarks. Our CcDPO not only improves performance in multi-image settings but also preserves strong capabilities on single-image tasks.

| Models | Parameter | MMStar | SQA | MMMU | POPE | HallBench | MMB | OCR | AI2D | Avg. |
|---|---|---|---|---|---|---|---|---|---|---|
| LLaVA-v1.6 [81] | 7B | 37.6 | 87.5 | 35.8 | 70.3 | 51.6 | 69.8 | 53.7 | 67.0 | 59.1 |
| Qwen-VL-Chat [82] | 7B | 34.5 | 68.8 | 35.9 | 74.9 | 39.2 | 61.8 | 48.8 | 63.0 | 53.3 |
| Idefics2 [20] | 8B | 49.5 | 88.7 | 43.0 | 86.2 | - | 75.7 | - | 72.3 | 69.2 |
| OpenFlamingo [83] | 9B | 36.9 | 44.8 | - | 52.6 | 38.4 | 32.4 | 14.9 | 31.7 | 35.9 |
| InstructBLIP [84] | 13B | 32.7 | 54.1 | - | 86.1 | 45.3 | 38.3 | 27.6 | 40.6 | 46.3 |
| Emu2-Chat [21] | 37B | 40.7 | 68.2 | 36.3 | 88.0 | - | 63.4 | 43.6 | 49.7 | 55.7 |
| LLaVA-OV [32] | 7B | 58.7 | 92.1 | **47.7** | 86.1 | 52.9 | 81.8 | 47.3 | 81.6 | 68.5 |
| + SFT | 7B | 57.8 | 91.5 | 47.1 | **88.4** | 57.2 | 81.5 | 50.2 | 81.6 | 69.4 |
| + MIA-DPO [27] | 7B | 57.4 | 92.4 | 45.1 | 87.9 | 55.4 | 80.9 | **52.1** | 81.5 | 69.0 |
| + CcDPO (Ours) | 7B | **59.5** | **92.6** | 45.7 | 86.6 | **58.4** | **81.9** | 51.0 | **82.1** | **69.7** |
| Qwen2-VL [33] | 7B | 57.8 | **84.1** | 50.6 | 85.9 | 66.9 | 81.2 | **85.6** | 78.9 | 73.8 |
| + SFT | 7B | 55.0 | 82.7 | 50.0 | 87.7 | 66.7 | 81.0 | 84.8 | 78.5 | 73.3 |
| + MIA-DPO [27] | 7B | 58.2 | 84.0 | 48.6 | **88.4** | 62.7 | 80.8 | 85.1 | 78.9 | 73.3 |
| + CcDPO (Ours) | 7B | **58.7** | 82.8 | **50.7** | 87.1 | **68.8** | **81.6** | 83.5 | **79.7** | **74.1** |

## 5.2 Main Results

**Results on General Multi-Image Tasks.** As shown in Table 2, we evaluate CcDPO across diverse multi-image benchmarks that span a wide range of reasoning skills. CcDPO consistently outperforms both the SFT baseline and other DPO-based methods on all datasets, with a notable gain of +4.8 points on BLINK, which focuses on multi-view and spatial reasoning. On the large-scale MuirBench dataset—where each sample contains an average of 4.3 images and up to 9 images—CcDPO achieves the largest improvement of +6.1 points, demonstrating its strength in modeling complex multi-image dependencies such as fine-grained perception, sequential cues, and holistic context. In contrast, MIA-DPO underperforms on MuirBench, highlighting its global context limitations. Consistent gains across LLaVA-OV and Qwen2-VL variants validate our approach's generality and effectiveness.

**Results on General Single-Image Tasks.** While previous works [27, 44] indicate that multi-image training can degrade single-image understanding, our CcDPO, in contrast, generally yields performance gains on most single-image datasets as shown in Table 3, averaging +1.2 points for LLaVA-OV and +0.3 for Qwen2-VL. Visually-driven tasks like HallBench exhibit the largest improvements, up to +5.5 points under CcDPO. Conversely, for tasks with relatively low reliance on visual information, exemplified by ScienceQA [75], our method showed no notable gains, and performance slightly declined. These results highlight CcDPO's robustness: it not only excels in multi-image scenarios but also preserves, and often enhances, single-image capabilities. We attribute this success to our

Table 4: **Performance on the image retrieval task from needle-in-a-haystack MM-NIAH [85].** We compare our proposed method, CcDPO, with DPO-based baselines across 1K–24K contexts, where the number of images ranges from a few to over a hundred. CcDPO consistently outperforms prior methods, demonstrating its strength in capturing fine-grained details in ultra-long image sequences.

| Models | Parameter | 1K | 2K | 4K | 8K | 12K | 16K | 24K | Average |
|---|---|---|---|---|---|---|---|---|---|
| LLaVA-OV-7B [32] | 7B | 89.2 | 88.1 | 82.3 | 71.2 | 65.0 | 60.9 | 45.0 | 71.7 |
| + SFT | 7B | 92.0 | 93.9 | 88.2 | 80.7 | 74.8 | 69.4 | 49.9 | 78.4 |
| + MIA-DPO [27] | 7B | 93.9 | 94.6 | 90.5 | 85.1 | 75.5 | 68.8 | 59.4 | 81.1 |
| + CcDPO (Ours) | 7B | **95.3** | **96.9** | **91.4** | **89.6** | **78.8** | **74.5** | **64.7** | **84.5** |
| Δ | - | **+6.3** | **+8.8** | **+9.1** | **+18.4** | **+13.8** | **+13.6** | **+19.7** | **+12.8** |

Table 5: **Ablation study of two-level CcDPO on MIBench and MIRB tasks** requiring perception, comparison, and reasoning across multiple images. Detailed task descriptions are in the Appendix B.

| Models | MIBench Benchmark | | | | | | | | MIRB Benchmark | | | |
|---|---|---|---|---|---|---|---|---|---|---|---|---|
| | GC | SD | TR | LR | FVR | TRI | VTK | TVK | Know. | Reas. | Perc. | M-Hop |
| LLaVA-OV-7B [32] | 87.7 | 85.9 | 72.6 | 74.5 | 96.5 | 77.5 | 42.7 | 67.1 | 70.0 | 44.0 | 50.0 | 12.0 |
| + Context-Level | 87.4 | 88.6 | 69.5 | 76.0 | 97.9 | 76.7 | 43.4 | 68.8 | 75.0 | 44.0 | 52.0 | 15.0 |
| ⊕ Needle-Level | 88.8 | 90.4 | 70.3 | 76.0 | 98.2 | 77.6 | 52.6 | 69.8 | 72.0 | 48.0 | 55.0 | 18.0 |
| Δ | +1.1 | +4.5 | -2.3 | +1.5 | +1.7 | +0.1 | +9.9 | +2.7 | +2.0 | +4.0 | +5.0 | +6.0 |
| + SFT | 87.9 | 88.2 | 71.0 | 75.5 | 87.8 | 77.4 | 38.5 | 68.3 | 75.0 | 43.0 | 53.0 | 9.0 |
| + MIA-DPO [27] | 85.8 | 87.2 | 63.4 | 67.9 | 94.9 | 67.6 | 42.0 | 59.6 | 73.0 | 50.0 | 44.0 | 11.0 |

preference data design, which employs structured, per-image descriptions, thereby fostering precise understanding of individual images even within multi-image contexts.

## 5.3 Ablation Studies

**Comparison with SFT Trained on MultiScope-42K.** As shown in Table 2, Table 3, Table 4, CcDPO outperforms SFT across all benchmarks, achieving +3.0 on multi-image tasks, +0.6 on single-image tests, and a significant +6.1 gain on MM-NIAH. By integrating negative samples into DPO, CcDPO enhances discrimination between accurate and hallucinated outputs, improving fine-grained detail recognition and long-range dependency modeling while maintaining single-image performance. This demonstrates both robust generalization and superior contextual understanding, with negative sample integration proving essential to its performance advantages.

**Superior Context Scaling for Fine-Grained Detail Capture.** As shown in Table 4, CcDPO significantly outperforms prior DPO-based methods and SFT baselines on the challenging MM-NIAH [85] needle-in-a-haystack image retrieval task. Remarkably, CcDPO achieves consistent improvements across all tested context lengths (1K–24K), with its largest gain (+19.7 points) occurring at the maximum 24K context length compared to the baseline model. This demonstrates CcDPO's exceptional ability to: (1) Scale effectively to ultra-long image sequences, and (2) Capture fine-grained visual details critical for discriminating subtle differences in large image collections. The +12.8-point average improvement underscores its robustness in handling large-scale multi-image contexts. The hierarchical preference optimization in CcDPO enables precise, context-aware understanding—an essential capability for processing extensive visual information.

**Effectiveness of Context-Level Optimization.** The Context-Level DPO enhances alignment between language responses and the holistic visual context across image sequences. As shown in Table 6, this yields significant improvements on sub-tasks requiring global reasoning, including: Diagram Understanding (+5.1), Image-Text Matching (+9.8), Similarity Matching (+16.4). These gains reflect the module's ability to: (1) Capture semantic relationships across images, and (2) Maintain coherent multi-image descriptions through consistent attribute attribution. Notably, Scene Understanding and Retrieval tasks also benefit from improved global alignment, confirming that such optimization effectively reduces context omission and conflation errors in complex visual sequences.

**Effectiveness of Needle-Level Optimization.** The Needle-Level DPO improves the model's ability to capture fine-grained visual cues by contrasting localized content. As shown in Table 5 and Table 6, this is especially effective for tasks requiring detailed comparisons across images. In MIBench, the VTK task—where the model must link information across image cells—shows a large gain of +9.9, demonstrating that our visual preference signals help focus on factual visual details. Similarly,

Table 6: **Ablation study of CcDPO on the MuirBench dataset** across all sub-datasets, demonstrating significant performance gains on most subsets. The symbol ⊕ stands for method superposition.

| Models | Overall | Action | Similarity | Cartoon | Counting | Diagram | Difference | Geographic | I-T Match | Ordering | Scene | Grounding | Retrieval |
|---|---|---|---|---|---|---|---|---|---|---|---|---|---|
| LLaVA-OV-7B [32] | 42.5 | 35.9 | 33.1 | 35.8 | 24.7 | 55.0 | 30.0 | 46.0 | 46.9 | 20.3 | 72.0 | 29.7 | 47.6 |
| + Context-Level | 44.8 | 36.6 | 30.3 | 37.1 | 35.5 | 57.0 | 37.9 | 41.0 | 48.3 | 15.6 | 65.1 | 28.3 | 48.3 |
| ⊕ Needle-Level-TDPO | 47.8 | 42.3 | 48.0 | 34.6 | 38.0 | 58.3 | 40.3 | 41.0 | 55.4 | 14.1 | 68.9 | 30.1 | 45.9 |
| ⊕ Needle-Level-VDPO | 48.6 | 44.5 | 49.5 | 33.3 | 39.3 | 60.1 | 39.1 | 41.0 | 56.7 | 13.0 | 70.4 | 28.6 | 46.6 |
| Δ | +6.1 | +8.6 | +16.4 | -2.5 | +14.6 | +5.1 | +9.1 | -5.0 | +9.8 | -7.3 | -1.6 | -1.1 | -1.0 |
| Qwen2-VL-7B [33] | 40.5 | 40.8 | 46.4 | 41.0 | 39.7 | 41.9 | 34.4 | 25.0 | 54.3 | 9.3 | 65.0 | 28.5 | 20.5 |
| + Context-Level | 42.1 | 41.0 | 45.0 | 42.3 | 39.8 | 46.2 | 37.9 | 22.0 | 56.9 | 14.1 | 65.1 | 27.4 | 21.0 |
| ⊕ Needle-Level-TDPO | 42.3 | 39.6 | 43.9 | 41.0 | 39.7 | 46.5 | 39.1 | 22.0 | 56.3 | 15.7 | 66.7 | 26.2 | 22.9 |
| ⊕ Needle-Level-VDPO | 44.8 | 43.9 | 46.9 | 38.5 | 39.9 | 53.8 | 37.9 | 20.0 | 61.4 | 18.8 | 69.9 | 23.8 | 22.9 |
| Δ | +4.3 | +3.1 | +0.5 | -2.5 | +0.2 | +11.9 | +3.5 | -5.0 | +7.1 | +9.5 | +4.9 | -4.7 | +2.4 |

Table 7: Ablation on component and training order.

| Configuration | MuirBench | Mantis | BLINK | Q-Bench2 |
|---|---|---|---|---|
| Baseline | 42.5 | 60.4 | 51.1 | 73.8 |
| Context-Level Only | 44.4 | 69.0 | 55.3 | 76.4 |
| Needle-Level Only | 46.3 | 65.0 | 54.3 | 75.5 |
| Needle → Context | 46.5 | 66.3 | 54.1 | **77.8** |
| Context → Needle | **48.6** | **69.6** | **55.9** | 77.6 |

Table 8: Ablation on visual prompt types.

| Bbox | Seg. | Point | Circle | BLINK | Q-Bench2 |
|---|---|---|---|---|---|
| | | ✔ | ✔ | 55.1 | 76.0 |
| | ✔ | | ✔ | 56.2 | 76.6 |
| | ✔ | ✔ | | **56.8** | 76.3 |
| ✔ | | | ✔ | 54.5 | **77.2** |
| ✔ | | ✔ | | 55.9 | 76.8 |

in MuirBench, we observe strong gains on Action Understanding (+8.6), Counting (+14.6), and Difference Spotting (+9.1), all of which depend on localized perception. These results indicate that Needle-Level DPO significantly boosts the model's perceptual grounding and resistance to detail-level hallucinations, complementing the context-level DPO for better multi-image understanding.

**Synergistic Two-Level Optimization in CcDPO.** As evidenced in Table 7, performing context-level DPO prior to needle-level DPO consistently yields optimal benchmark performance, underscoring a hierarchical interdependence between the two stages. This workflow mirrors human general reasoning for multi-image problems: one first understands the global context, then zooms in to focus on specific details as needed for the task. Notably, inverting the training sequence or isolating either stage results in suboptimal outcomes, further validating the necessity of our two-stage sequential optimization.

**Sensitivity Analysis of Visual Prompt.** As shown in Table 8, our ablation studies reveal minimal performance variation (±1 point) across visual prompt types (bounding boxes, circles, segmentation masks) during the Needle-Level Language-based Preference Optimization stage, underscoring our method's robustness. Notably, we observe benchmark-specific patterns: fine-grained prompts (e.g., segmentation masks) excel in spatial reasoning tasks BLINK by capturing precise local details, while coarse-grained prompts (e.g., bounding boxes) perform better in image-level comparisons Q-Bench2 through efficient spatial guidance. Consequently, we adopt bounding box + keypoint as the final format, balancing accuracy and annotation practicality optimally.

**Limitations.** While CcDPO is primarily designed for general multi-image reasoning, it does not explicitly model temporal dependencies, which may limit its performance on video-like inputs. However, our framework is readily extensible to such video data by incorporating video clip descriptions (as demonstrated in Table 13). Similarly, although OCR supervision is limited in our current dataset, CcDPO can be naturally enhanced with targeted text-centric preference data in future work.

# 6 Conclusion

This work introduces **CcDPO**, a two-level preference optimization method for enhancing multi-image understanding in MLLMs. By decoupling learning into context-level and needle-level stages, CcDPO addresses key hallucination issues including context omission, conflation, and detail misinterpretation. The context-level module promotes holistic sequence comprehension via structured caption preferences, while the needle-level module strengthens fine-grained perception through visual prompts and contrastive supervision. To support optimization, we construct **MultiScope-42k**, a large-scale dataset with automatically generated multi-level preference pairs. Experiments across seven multi-image benchmarks show that CcDPO achieves consistent improvements over SFT and prior DPO variants, confirming its effectiveness in aligning MLLMs with both global and local visual content.

## Acknowledgments

This work was supported by the National Science Fund for Distinguished Young Scholars (No.62025603), the National Natural Science Foundation of China (No. 62576299, No. 625B2156, No. U21B2037, No. U22B2051, No. U23A20383, No. U21A20472, No. 62176222, No. 62176223, No. 62176226, No. 62072386, No. 62072387, No. 62072389, No. 62002305, and No. 62272401), the Natural Science Foundation of Fujian Province of China (No. 2021J06003, No. 2022J06001), and the Fundamental Research Funds for the Central Universities.

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

# Technical Appendices

In this appendix, we provide additional materials to support a more comprehensive understanding of our proposed method and dataset. **In Sec. A**, we detail the low-cost construction pipeline of **MultiScope-42k** and conduct comparative data analysis with MIA-DPO, including token length distributions, word cloud statistics, and supervision source breakdown. We also clarify the image source overlap between training data and benchmarks to ensure fair evaluation. **In Sec. B**, we summarize all benchmarks used in evaluation, including seven multi-image and eight single-image benchmarks. **In Sec. C**, we provide additional experimental results, including ablation studies on training data volume and supervision granularity to assess their impact on model performance. **In Sec. D**, we present qualitative observations and visualizations of preference pairs.

## A  MultiScope-42k: A Context-to-Cue Captioning DPO Dataset

### A.1  Low-cost Question-Answer Pair Construction

Constructing high-quality instruction-response preference pairs for multi-image learning traditionally requires extensive manual annotation, especially when capturing subtle context dynamics or region-level semantics. To address this bottleneck, we design a low-cost, fully automated pipeline for question-answer pair construction, enabling efficient and scalable data generation with broad coverage and controlled distributional properties.

**Automated Caption Pool Sampling.** We first leverage existing vision-language datasets—LLaVA-23K [61], MDVP [63], and MVC [49]—to construct a diverse caption pool, containing both image-level and region-level descriptions. By decoupling question construction from caption generation, we are able to sample visual contexts and their aligned captions independently, facilitating large-scale composition of input-output examples.

**Structured QA Formatting.** Given a sampled image or image sequence, we construct templated instructions (e.g., "Describe the following images" or "Please describe the marked area in each image") to form queries. For the corresponding answers, we use structured formats that encourage compositional reasoning and grounding, such as:

```
[ For Image 1:  <caption 1>, For Image 2:  <caption 2>, ...]

    [ For the marked area of Image X: <caption X>, ...]
```

This approach allows flexible variation in image number, visual scope, and response granularity—supporting both context- and region-level supervision.

**Controlled Perturbation for Preference Learning.** To generate preference pairs without additional labeling, we apply lightweight perturbation strategies to the answer side only. Specifically:
• (1) Truncation and swapping: simulate omissions and misalignments in context-level answers.
• (2) Region mismatches: in needle-level samples simulate detail hallucination.
These perturbations require **no human involvement** yet introduce controlled errors mirroring real-world MLLM failure modes, enabling scalable preference pair generation with low cost.

**Efficient Coverage of Diverse Visual Distributions.** With stratified control over image domains, scene compositions, and region attributes via automated sampling, we construct MultiScope-42k: a large-scale corpus covering diverse multi-image tasks. Its diversity in visual layout and semantic granularity delivers robust preference supervision across image types and reasoning levels.

Overall, our pair construction strategy eliminates the need for dense manual annotation while producing rich and challenging preference data at scale—offering a practical solution for instruction tuning in multi-image multimodal models.

### A.2  Data Analysis

To better understand the characteristics of our preference supervision, we conduct a comparative analysis of MultiScope-42k and MIA-DPO from both lexical and structural perspectives.

**Dataset Composition Overview.** Summary statistics of the proposed MultiScope-42k are presented in Table 9, which is divided into three core subsets by supervision granularity and image source, totaling 41.8k preference pairs (27.3k context-level, 10.8k needle-level TDPO, and 3.7k needle-level

Table 9: Summary statistics of MultiScope-42k by supervision level and image source, including total pairs, number of images per instance, and average token lengths of chosen and rejected responses.

| Level | Image Source | Total | Images Range | Avg. Chosen Len. | Avg. Rejected Len. |
|---|---|---|---|---|---|
| Context-Level | COCO-2014 | 27.3k | [2, 5] | 285.24 | 165.45 |
| Needle-Level-TDPO | COCO-2017 | 10.8k | [2, 4] | 173.35 | 173.97 |
| Needle-Level-VDPO | Flickr30k | 3.7k | [2, 4] | 66.09 | 65.98 |

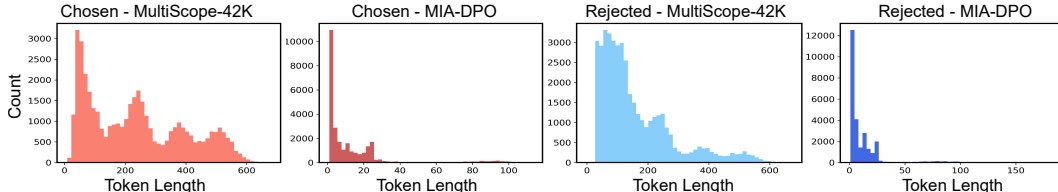

Figure 4: Token length distributions of chosen and rejected responses in our MultiScope-42k and MIA-DPO [27]. MultiScope-42k exhibits significantly longer and more diverse answers, while MIA-DPO responses remain short and concentrated, indicating a simpler response pattern.

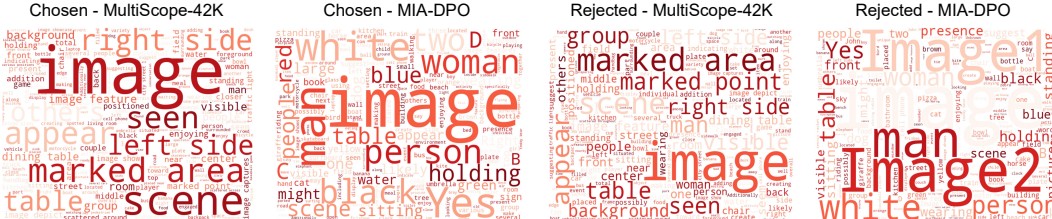

Figure 5: The dataset word cloud comparison between our MultiScope-42k and MIA-DPO [27].

VDPO) to cover supervision needs of varying granularities in multi-image scenarios. The 27.3k context-level pairs incorporate approximately 7k multi-image QA data from Mantis [44] to enhance the model's instruction-following capability. The negative samples are generated by the model based on responses to images with Gaussian noise added, inducing the model to produce answers that are unfaithful to the original images. Regarding the number of images per instance, all subsets contain 2 to 5 images (up to 5 for context-level, 4 for needle-level), aligning with typical multi-image learning settings and supporting models in modeling multi-image sequence contexts.

**Token Length Distributions.** Fig. 4 presents the token length histograms of both *chosen* and *rejected* responses. MultiScope-42k responses are significantly longer on average and display a wider spread, with many responses exceeding 400 tokens. This reflects the dataset's multi-stage, image-wise captioning format and compositional design. In contrast, MIA-DPO responses are short and concentrated, with the majority under 30 tokens. This further suggests that MultiScope-42k provides richer and more diverse supervision signals, especially for multi-image reasoning.

**Linguistic Focus via Word Cloud.** We visualize the answer sets of both datasets using word clouds. As shown in Fig. 5, MultiScope-42k responses prominently feature structured and spatially grounded expressions such as *"image," "marked area," "left," "right," "foreground," "scene"*, indicating a strong alignment with multi-image, region-specific prompts. In contrast, MIA-DPO emphasizes atomic visual concepts (e.g., *"man," "table," "dog," "red"*), which are well-suited for single-image tasks but lack explicit inter-image reference or structural composition.

### A.3 Dataset Independence Statement

To ensure fair and unbiased evaluation, we verify that the benchmarks used for testing do not overlap with the data sources involved in model training. We employ three types of annotated data for two-level DPO training. (1) Image-level captions are sourced from **LLaVA-23K** [61], which is derived from the COCO-2014 dataset. (2) Region-level captions come from the **MDVP** dataset [63], based on COCO-2017. (3) Visual contrastive preference pairs are taken from **MVC** [49], whose image sources include **CounterCurate** [86] and **FineCops-Ref** [87].

We conducted a thorough review of all evaluation benchmarks for potential overlap with these training sources (COCO-2014, COCO-2017, Flickr30k). While most multi-image benchmarks appear to be independently constructed, we conservatively flag the following for partial or uncertain overlap:

**MIRB** [68] explicitly uses COCO images (e.g., for counting), and includes content from ImageNet-R, OpenFoodFact, Bitton et al., and arXiv. **MIBench** [72] comprises multiple public datasets, including VrR-VG, which inherits COCO images via Visual Genome. **MUIRBench** [42] integrates existing (e.g., SeedBench, IconQA), derived (e.g., NLVR2, MMBench), and newly collected datasets. While its new data is COCO-free, MMBench and IconQA are known to include COCO images.

A summary of benchmark image sources is provided below:

**MUIRBench** combines new and repurposed datasets; some derived components include COCO. **MIRB** includes COCO, ImageNet-R, and other web-sourced content. **BLINK** uses synthetic and controlled real-world images. **Mantis-Eval** draws from web queries and manual composition, with no COCO usage. **NLVR2** uses Flickr images and is COCO-independent. **Q-Bench2** is based on IQA datasets like KonIQ-10k and BID, unrelated to COCO or Flickr30k. **MIBench** includes VrR-VG, which partially overlaps with COCO.

While a few benchmarks partially overlap with COCO or Flickr30k, evaluation fairness remains uncompromised for two reasons. First, our primary baseline MIA-DPO is also trained on LLaVA-23K (COCO-based), ensuring all methods are compared on equal footing. Second, our DPO training focuses on image-caption preference alignment, not question answering. These points validate the integrity and fairness of our evaluation protocol, despite partial dataset overlaps.

# B  Benchmark Sources

## B.1  More Details on the Construction of Context-AMBER-1K

We design a caption generation task as a proxy to systematically evaluate MLLMs' multi-image understanding, identifying three fundamental hallucination types—*Context Omission*, *Context Conflation*, and *Detail Misinterpretation*—that critically degrade model performance. To enable this evaluation, we construct **Context-AMBER-1K** by systematically concatenating images from the single-image AMBER dataset [37] into sequences of two types: short-context sequences with 4 images and long-context sequences with 8 images. Each input is paired with the prompt: ``Please sequentially describe each of the images shown above. Use the format:  For Image *:<description>.''.

To evaluate **Detail Misinterpretation**, we employ three established metrics: (a) CHAIR [36]; (b) Response-level hallucination rate (Hal); and (c) Cognition-based hallucination (Cog). Each generated caption is matched to its corresponding image using regular expressions to extract per-image descriptions. We then compute hallucination scores for each image separately and average these scores across the entire image sequence. To assess **Context Omission** and **Context Conflation**, we introduce two rule-based penalization strategies:

(1) For context omission, if fewer than $N$ captions are generated (e.g., the response ends prematurely at {For Image N-1:  <caption N-1>}), we explicitly pad the output to {For Image N: <caption N-1>}) to maintain a consistent structure. Missing or incomplete captions are heavily penalized. Additionally, we introduce the (d) Sequence Coverage Rate (SCover), a metric that evaluates caption completeness across image sequences and quantifies the degree of context omission.

(2) For context conflation, we include the keyword ``sequentially'' in the prompt to encourage models to describe images in order. If the model cannot clearly associate each description with its corresponding image (e.g., For Image 1:<caption1>, For Image 3:<caption2>, For Image 2:<caption3>), the resulting hallucination scores will be higher for out-of-order descriptions compared to their correct GTs, as the mismatch leads to a larger discrepancy.

## B.2  Multi-Image Benchmarks

We employ seven multi-image benchmarks—MUIRBench [42], MIRB [68], BLINK [69], Mantis-Eval [44], NLVR2 [70], Q-Bench2 [71], and MIBench [72]—to holistically evaluate multi-image

reasoning across four key dimensions: co-reference alignment, fine-grained comparison, contextual reasoning, and temporal understanding.

**MUIRBench** [42] is a comprehensive benchmark specifically designed to evaluate the robustness of multimodal large language models (MLLMs) in multi-image understanding scenarios. It comprises 2,600 multiple-choice questions and 11,264 images, averaging 4.3 images per instance. The benchmark covers 12 distinct multi-image understanding tasks—including action understanding, diagram reasoning, geographic comprehension, and visual retrieval—spanning 10 diverse multi-image relation types such as temporal, narrative, and scene-multiview relations. To ensure both comprehensiveness and robustness, MUIRBench adopts a pairwise design: each standard (answerable) question is paired with an unanswerable variant with minimal semantic perturbations. This enables fine-grained assessments of both reasoning capability and abstention behavior.

**MIRB** [68] is a comprehensive benchmark designed to evaluate vision-language models (VLMs) on four distinct aspects of multi-image understanding: perception, visual world knowledge, reasoning, and multi-hop reasoning. It comprises 925 multi-image questions across these categories, averaging 3.78 images per question, with some tasks requiring up to 42 images for complex reasoning. Unlike prior benchmarks that reuse video frames, MIRB independently sources images from real-world domains, such as code snippets, sightseeing scenes, food ingredient lists, and arXiv papers, ensuring diverse and challenging visual contexts. The benchmark includes a wide array of tasks: image jigsaw reconstruction, object counting, attribute matching (**Perception**); food label comparison and geographic recognition (**Knowledge**); visual analogy, code understanding, 3D scene analysis (**Reasoning**); and synthetic logic chains and citation lookups (**Multi-Hop**). Each question is formulated to necessitate reasoning across multiple images rather than from a single image.

**MIBench** [72] is a large-scale benchmark designed to comprehensively evaluate the fine-grained multi-image understanding abilities of multimodal large language models (MLLMs). It categorizes multi-image inputs into three representative scenarios—Multi-Image Instruction (MII), Multimodal Knowledge-Seeking (MKS), and Multimodal In-Context Learning (MIC)—covering a total of 13 distinct tasks and 13,000 annotated samples. In the MII setting, the model must perform perception, comparison, and reasoning over multiple images across five tasks: general comparison (**GC**), subtle difference (**SD**), visual referring (**VR**), temporal reasoning (**TR**), and logical reasoning (**LR**). The MKS scenario evaluates the model's ability to extract and align information from interleaved image-text knowledge sources through four tasks: fine-grained visual recognition (**FVR**), text-rich image VQA (**TRI**), vision-linked textual knowledge (**VTK**), and text-linked visual knowledge (**TVK**). Finally, the MIC setting assesses multimodal in-context learning across four tasks, including close-ended and open-ended VQA, hallucination mitigation, and demo-based task learning.

**BLINK** [69] tests rapid visual cognition through perceptual similarity, forensic analysis, and spatiotemporal matching. It includes tightly-controlled multi-image tasks such as depth estimation, object matching, and outlier detection, with an emphasis on speed and perceptual accuracy.

**Mantis-Eval** [44] introduces 217 multi-image tasks curated for conceptual inference, including abstract reasoning over physical quantities such as number, size, and weight. It combines both multiple-choice and open-ended questions, drawing from web-sourced image sets manually organized into logical visual groupings.

**NLVR2** [70] (Natural Language Visual Reasoning) assesses a model's ability to verify textual hypotheses against a pair of images. Each sample requires binary classification (True/False) over whether the provided statement is consistent with both images, making it a canonical test for visual entailment and compositional reasoning.

**Q-Bench2** [71] is a diagnostic benchmark tailored for evaluating visual quality perception and comparative assessment across image sets. It challenges models to identify subtle visual artifacts, distortions, or improvements between similar images. Our evaluation is based on the Q-Bench2-A1-dev subset, which emphasizes multi-image multiple-choice assessments for perceptual judgment.

### B.3 Single-Image Benchmarks

We test the model on eight representative single-image benchmarks assess specific multimodal capabilities: (1) Academic/Scientific Reasoning: MMMU [73], MMStar [74], ScienceQA [75], (2) Diagram Understanding: AI2D [76], (3) Robustness against hallucinations: POPE [77], HallBench [38], (4) General Multimodal Abilities: MMBench [78], (5) Text Recognition: OCRBench [79]. The results

Table 10: **Benchmark Sources.** We have included detailed information for all the multi-image and single-image benchmarks tested in the paper in the table.

| Setting | Models | Evaluation Metric | Number | Source |
|---|---|---|---|---|
| **Multi-Image Benchmark** | MUIRBench [42] | Multiple Choice | 2,600 | **MUIRBench** |
| | MIRB [68] | Multiple Choice | 925 | **MIRB** |
| | MIBench [72] | Multiple Choice | 13,000 | **MIBench** |
| | BLINK [69] | Multiple Choice | 3,807 | **BLINK** |
| | NLVR2 [70] | Multiple Choice | 6,967 | **NLVR2** |
| | Q-Bench2 [71] | Multiple Choice | 1,000 | **Q-Bench2** |
| | Mantis-Eval [44] | Multiple Choice | 217 | **Mantis-Eval** |
| **Single-Image Benchmark** | MMStar [74] | Multiple Choice | 1,500 | **MMStar** |
| | MMMU [73] | Multiple Choice | 1,050 | **MMMU** |
| | Sci-QA [75] | Multiple Choice | 4,241 | **ScienceQA** |
| | POPE [77] | Yes/No | 9,000 | **POPE** |
| | HallBench [38] | Yes/No | 951 | **HallusionBench** |
| | MMB [78] | Multiple Choice | 1,164 | **MMBench** |
| | OCR [79] | VQA | 1,000 | **OCRBench** |
| | AI2D [76] | Multiple Choice | 3,090 | **AI2D** |

on this diverse set of benchmarks demonstrate the effectiveness of the proposed method, particularly in multi-image settings, confirming significant performance improvements.

**MMMU** [73] (Massive Multimodal Multitask Understanding) includes over 10k university-level questions from 30+ disciplines such as physics, medicine, and art. It requires detailed reasoning over image-text inputs and is designed to evaluate advanced academic-level understanding.

**MMStar** [74] is a comprehensive diagnostic benchmark covering various sub-tasks such as OCR, VQA, and caption grounding, offering structured and hierarchical annotations across domains like natural science, medicine, and design.

**ScienceQA** [75] contains over 21k science questions aligned with elementary and middle school curricula, involving images such as diagrams and charts. It tests the model's capability to perform science-related visual reasoning in a multimodal format.

**AI2D** [76] (Allen Institute Diagram) features manually annotated science diagrams with associated multiple-choice questions. It focuses on assessing the model's understanding of labeled structures and their functional roles within the image.

**POPE** [77] (Position and Object-level Prompt Evaluation) is designed to test a model's resistance to hallucinations. It uses minimally perturbed prompts to identify failure cases in positional grounding and object identification, highlighting model robustness.

**HallBench** [38] provides a structured framework to measure hallucination frequency and grounding quality by comparing model outputs with annotated ground truths. It supports fine-grained scoring across categories such as incorrect object mentions or unsupported claims.

**MMBench** [78] is a general-purpose evaluation benchmark comprising questions across 11 modalities including VQA, captioning, OCR, and commonsense reasoning. It uses GPT-4-based grading to ensure high-fidelity evaluation of answer correctness.

**OCRBench** [79] specifically targets the model's capability to recognize and reason about text in the visual domain, covering a range of document layouts, fonts, and multilingual content with both exact-match and reasoning-based questions.

## C   More Experiments

**Data Scale Alignment with MIA-DPO.** To assess the impact of training size and ensure a fair comparison with MIA-DPO, we conduct an ablation in Tab. 11 using a similar total number of preference pairs. Specifically, we randomly sample 13.6k from our 27.3k Context-Level pairs and combine them with the fixed 14.5k Needle-Level data, resulting in a 28.1k training set—comparable to MIA-DPO's 28.9k. Notably, under this matched training data size, our **CcDPO** still outperforms

Table 11: **Ablation on training data volume.** To match MIA-DPO's training data size, we down-sample our Context-Level data to 13.6k while keeping needle-level data fixed. The results reveal a trade-off between modeling global context and capturing fine-grained details: while reduced Context-Level data leads to performance drops on most multi-image tasks, benchmarks like BLINK and Q-Bench2—focused on localized perception—benefit from a higher proportion of needle-level data.

| Models | Data Size | MuirBench | MIRB | BLINK | Mantis | NLVR2 | MIBench | Q-Bench2 | Average |
|---|---|---|---|---|---|---|---|---|---|
| LLaVA-OV [32] | - | 42.5 | 47.3 | 51.1 | 60.4 | 89.4 | 73.6 | 73.8 | 62.5 |
| + MIA-DPO [27] | 28.9K | 41.4 | 48.0 | 53.7 | 60.3 | 88.2 | 67.8 | 74.0 | 61.9 |
| + CcDPO (Ours) | 28.1K | 46.7 | 51.2 | **56.5** | 69.1 | 90.7 | 72.1 | **79.3** | 66.5 |
| + CcDPO (Ours) | 41.8K | **48.6** | **51.4** | 55.9 | **69.6** | **91.2** | **75.2** | 77.6 | **67.1** |
| Δ | - | +6.1 | +4.1 | +4.8 | +9.2 | +1.8 | +1.6 | +3.8 | +4.6 |

Table 13: **Performance on VideoMME Benchmark.** CcDPO-Video improves baseline performance across core video tasks, verifying cross-domain generalizability.

| Model | Overall | Perception | | | | | OCR Problems |
|---|---|---|---|---|---|---|---|
| | | Temporal | Spatial | Attribute | Action | Object | |
| LLaVA-OV [32] | 53.7 | 0.491 | 0.556 | 0.685 | 0.505 | 0.559 | **0.568** |
| + CcDPO-Video (Ours) | **54.4** | **0.509** | **0.574** | **0.698** | **0.527** | **0.582** | 0.547 |

| Model | — | Reasoning | | | | | Information Synopsis |
|---|---|---|---|---|---|---|---|
| | | Counting | Temporal | Spatial | Action | Object | |
| LLaVA-OV [32] | — | 0.354 | 0.384 | 0.714 | **0.523** | **0.518** | **0.675** |
| + CcDPO-Video (Ours) | — | **0.377** | **0.401** | **0.732** | 0.519 | 0.511 | 0.663 |

MIA-DPO across all benchmarks, demonstrating the effectiveness of our structured, dual-level supervision. In particular, the reduced-context setting yields better performance on BLINK and Q-Bench2, suggesting that a relatively higher proportion of needle-level data may benefit fine-grained perceptual tasks. On the other hand, performance on context-heavy benchmarks drops slightly, likely due to weaker global context modeling. Overall, training with the full 42k preference set (28.1k Context-Level + 14.5k Needle-Level) leads to the best average performance. These results highlight the advantage of high-quality, large-scale supervision, while also revealing a trade-off between contextual alignment and perceptual precision.

**Ablation of Training Strategies.** We investigate whether our multi-stage training strategy—first training on Context-Level data, followed by Needle-Level supervision—is more effective than a single-stage approach that mixes both types of data from the beginning. As shown in Tab. 12, the multi-stage strategy consistently outperforms the mixed-data alternative across benchmarks. We attribute this improvement to the sequential learning structure. In the first stage, the model learns to capture global context and image-level coherence through structured, sequence-aligned supervision. Once this foundation is established, the second stage refines the model's ability to attend to local, fine-grained visual cues via region-specific preference optimization. In contrast, the one-stage strategy may dilute the model's focus by simultaneously exposing it to competing global and local objectives, making optimization less efficient. These results suggest that decoupling context modeling and fine-grained grounding into separate stages can better guide the model toward hierarchical visual reasoning.

Table 12: **Ablation on training strategy.** Multi-stage training outperforms one-stage mixed training.

| Strategy | MuirBench | BLINK | Mantis |
|---|---|---|---|
| LLaVA-OV | 42.5 | 51.1 | 60.4 |
| One-Stage | 46.2 | 55.1 | 65.4 |
| Multi-Stage | **48.5** | **55.9** | **69.6** |

**Framework Extension to Video Data.** To explore our framework's potential for sequential visual data beyond multi-images, we extend its core idea of stitching individual images into multi-image sequences to video scenarios, adopting the structured input format: $\langle$video clip$_1\rangle$ $\langle$caption$_1\rangle$ $\langle$video clip$_2\rangle$ $\langle$caption$_2\rangle$. We directly adapt our **Sequence Truncation** and **Content Swapping** strategies for negative sample generation, avoiding costly manual re-labeling or rejection sampling. Following this pipeline, we construct a 7k-scale video preference dataset and perform DPO training on the LLaVA-OV baseline (denoted as **CcDPO-Video**).

As shown in Table 13, CcDPO-Video achieves a **0.7-point overall gain** over LLaVA-OV on VideoMME, with measurable improvements in temporal perception (+0.018), action recognition (+0.022), and object recognition (+0.023). These exploratory results highlight two promising strengths of our approach: (1) Video clip stitching—an extension of our multi-image sequence construction paradigm—is **computationally economical**, obviating the need for manual re-annotation or heavy computational overhead associated with existing methods; (2) Its perturbation-based sampling is **transferable** to sequential visual data: positive samples use semantically ordered video clips, while negatives are generated via scrambling or truncation. This trial thus demonstrates a feasible pathway to extend multi-image DPO frameworks to video understanding tasks.

## D    More Observations

As illustrated in Fig. 6, 8, 7, 10, and 11, we present additional qualitative examples of the constructed preference pairs used in our CcDPO training. These examples cover a range of perturbation types across both Context-Level and Needle-Level supervision, each designed to target specific failure modes in multi-image reasoning.

**Fig. 6** shows a context-level *complete content omission* scenario, where the rejected response omits part of the image sequence. This simulates a context omission error, encouraging the model to favor complete and coherent descriptions across multiple images.

**Fig. 7** demonstrates a context-level *partial content omission* scenario, where the rejected response uses overly brief or generic captions. Such examples train the model to prefer rich, fine-grained visual descriptions that capture meaningful details.

**Fig. 8** presents a case of *caption swapping*, where descriptions are mismatched with image indices. This context conflation perturbation helps the model learn to correctly associate content with its corresponding image.

**Fig. 9 and Fig. 10** move to the Needle-Level and show a case where region-specific (bbox or point) captions are mismatched. This encourages the model to distinguish between accurate and inaccurate local grounding within a given image.

**Fig. 11** illustrates the use of contrastive image inputs for the same caption, where one image visually aligns with the text while the other introduces a subtle contradiction. This visual preference pair sharpens the model's ability to resolve ambiguous references through visual evidence alone.

Together, these examples highlight how different types of preference signals are systematically constructed to improve the model's ability to reason over sequences, resolve visual references, and discriminate between subtle semantic mismatches. By explicitly encoding both global and local reasoning objectives into the supervision data, CcDPO promotes more robust and interpretable multi-image alignment.

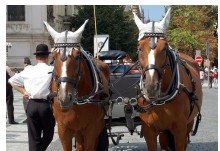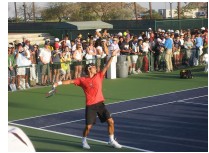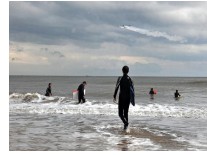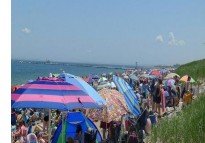

**Question: What is this photo about?<image><image><image><image>**

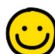 **Chosen:** **For image 1:** The image depicts two horses pulling a carriage on a street. Both horses are beautifully decorated, and one of them appears to be wearing chain mail hats. They are tethered to the carriage and working in unison to transport it through the street.\n\nThere are several people in the scene, all around the horses and carriage. Some of the people are walking along the sidewalk, while others may be getting on or off the carriage. A stop sign is located on the ride side of the scene, indicating a controlled intersection. \n

**For image 2:** The scene shows a man on a tennis court ready to serve the tennis ball. He is holding a tennis racket in his hand, with the ball positioned above him, preparing for the hit. Several spectators surround the court area, attentively watching the man's performance.\n\nOther items can be seen around the court, such as a suitcase located on the right side and a backpack towards the left. Many of the spectators are focused on the tennis player, eagerly anticipating the start of the game. \n

**For image 3:** The image shows a group of people wearing wet suits at a beach, all carrying surfboards under their arms while in the water. Some surfers are farther out in the water, while one person can be seen walking towards the water with their board. \n\nIn the background, multiple airplanes are flying in the sky, and a small boat is noticeable near the middle of the scene. The bodyboarders seem to be having a great time enjoying their water activities, with the presence of clouds contributing to the beach atmosphere. \n

**For image 4:** The image is of a crowded beach on a clear day with a blue sky. There are numerous people enjoying their time by the ocean, some standing or walking around, while others have set up colorful umbrellas and chairs to relax on the beach. \n\nMany large umbrellas are spread along the shoreline near the water, providing shade for beachgoers, and brightening up the scene with their vibrant colors. A variety of chairs can be found in the area, some near the umbrellas and others scattered across the beach. \n\nA few surfboards and handbags are also visible, indicating that people are likely engaging in water activities and have brought personal belongings for their day at the beach.

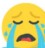 **Rejected:** **For image 1:** The image depicts two horses pulling a carriage on a street. Both horses are beautifully decorated, and one of them appears to be wearing chain mail hats. They are tethered to the carriage and working in unison to transport it through the street.\n\nThere are several people in the scene, all around the horses and carriage. Some of the people are walking along the sidewalk, while others may be getting on or off the carriage. A stop sign is located on the ride side of the scene, indicating a controlled intersection. \n

**For image 4:** The image is of a crowded beach on a clear day with a blue sky. There are numerous people enjoying their time by the ocean, some standing or walking around, while others have set up colorful umbrellas and chairs to relax on the beach. \n\nMany large umbrellas are spread along the shoreline near the water, providing shade for beachgoers, and brightening up the scene with their vibrant colors. A variety of chairs can be found in the area, some near the umbrellas and others scattered across the beach. \n\nA few surfboards and handbags are also visible, indicating that people are likely engaging in water activities and have brought personal belongings for their day at the beach.

Figure 6: Example of Context-Level preference pair with truncation perturbation (Context Omission).

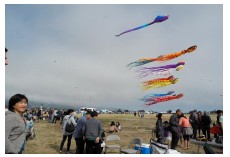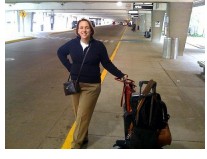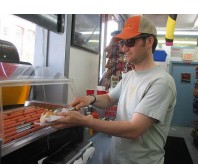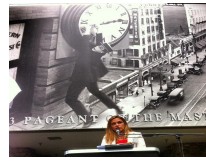

**Question: What is this photo about?<image><image><image><image>**

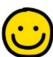 **Chosen: For image 1:** The image features a large group of people gathered at a sandy beach, flying various kites in the cloudy sky. There are more than ten people visible in the scene, dispersed throughout the beach area. The kites come in assorted shapes and sizes, with some resembling squid and octopus wind streamers soaring above the crowd. \n\nThere are also multiple chairs placed in the scene, accommodating those who want to take a break from kite flying or simply enjoy the spectacle. Additionally, a few handbags are lying on the beach, close to their respective owners. \n

**For image 2:** In the image, a woman is standing outside near a curb with her luggage. She has her hand on her hip, looking confident or perhaps waiting for someone. Her luggage consists of a suitcase next to her and a handbag placed closer to her. \n\nThe scene appears to be near an airport or a drop-off/pick-up area, with several cars parked or driving nearby. A bench is located on the far right of the scene, commonly found at such locations. Additionally, there are multiple other people and various sized handbags and suitcases spotted in the area, indicating that this is a bustling and busy place for travelers. \n

**For image 3:** The image features a man inside a convenience store or restaurant, in the process of placing a hot dog in a bun. He is surrounded by an assortment of hot dogs in various positions, indicating the convenience store or restaurant has a wide selection of hot dogs for customers to choose from. The man seems focused on preparing his chosen hot dog for consumption. \n

**For image 4:** A woman is standing on stage, addressing her audience with a microphone. She is wearing a white and red outfit, and there is a large black and white picture in the background. The woman appears to be giving a lecture or making a presentation to her students or a diverse audience.\n\nThere are several people scattered throughout the image, indicating a sizeable gathering listening to the woman's lecture. In the vicinity, various cars are parked outside the venue, some with a bit of distance between them. There is even a truck parked further out near some other vehicles. Additionally, there is a clock visible high up on a wall.",

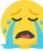 **Rejected: For image 1:** Crowds of people are at the beach with squid and octopus wind streamers above.

**For image 2:** A woman standing at an airport next to luggage.

**For image 3:** The man is at the convenience store selecting a hotdog.

**For image 4:** The woman is speaking to her audience on stage. ",

Figure 7: Example of Context-Level preference pair with caption shortening perturbation (Context Omission).

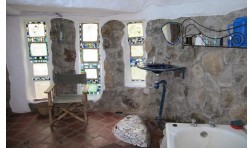 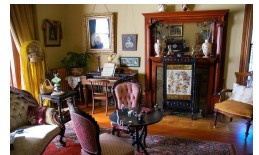 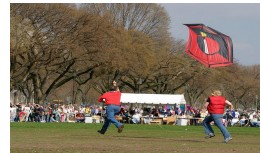

**Question: What is this photo about?<image><image><image>**

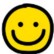 **Chosen:** **For image 1:** The image showcases a bathroom illuminated by a stained glass window, with a white bathtub located next to a stone wall. The room features stone walls and floors, giving the bathroom a rustic feel.\n\nIn the bathroom, a sink is situated along the wall, while a wooden chair is placed in the foreground. Several bottles and cups can also be found near the sink, likely containing bathroom essentials, such as shampoo, soap, and other toiletries. \n

**For image 2:** The living room is elegantly arranged with old-fashioned, antique furniture and features exquisite details in the woodwork. There is an ornate couch situated on the left side of the room, and several antique style chairs are placed around a dining table. The parlor features an oriental rug, adding an extra layer of charm to the room.\n\nThe living room is also adorned with various decorations such as multiple vases and a potted plant, which accentuate the antique atmosphere.

**For image 3:** In the image, a group of people, including both men and women, is gathered in a park flying kites. Among them, two people are prominently seen flying a red kite in the field, possibly running to keep it airborne. The audience of onlookers is spread out in the park, watching and enjoying the kite-flying activities.\n\nSome chairs are placed around the park for people to sit and relax. Additionally, there are a few traffic lights visible, indicating that the park is likely situated near a road or a busy area.

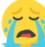 **Rejected:** **For image 1:** The image showcases a bathroom illuminated by a stained glass window, with a white bathtub located next to a stone wall. The room features stone walls and floors, giving the bathroom a rustic feel.\n\nIn the bathroom, a sink is situated along the wall, while a wooden chair is placed in the foreground. Several bottles and cups can also be found near the sink, likely containing bathroom essentials, such as shampoo, soap, and other toiletries. \n

**For image 2:** In the image, a group of people, including both men and women, is gathered in a park flying kites. Among them, two people are prominently seen flying a red kite in the field, possibly running to keep it airborne. The audience of onlookers is spread out in the park, watching and enjoying the kite-flying activities.\n\nSome chairs are placed around the park for people to sit and relax. Additionally, there are a few traffic lights visible, indicating that the park is likely situated near a road or a busy area.

**For image 3:** The living room is elegantly arranged with old-fashioned, antique furniture and features exquisite details in the woodwork. There is an ornate couch situated on the left side of the room, and several antique style chairs are placed around a dining table. The parlor features an oriental rug, adding an extra layer of charm to the room.\n\nThe living room is also adorned with various decorations such as multiple vases and a potted plant, which accentuate the antique atmosphere.

Figure 8: Example of Context-Level preference pair with swapping perturbation (Context Conflation).

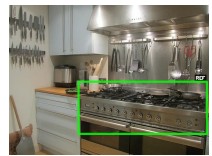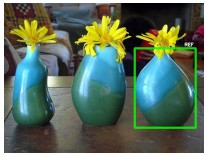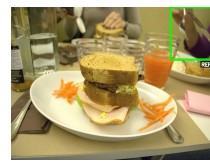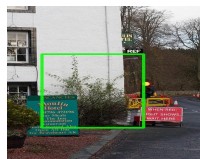

**Question:** <image><image><image><image>
Please provide a detailed description of the marked areas in each image. The marked areas are indicated by a green rectangle with a 'REF' label around it.",

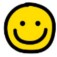

**Chosen: For the marked areas of image 1:** A large commercial-grade stove crawls across a significant section of the kitchen, sparking with arrays of burners. Numerous knobs dot the front, lending a look of complexity and functionality to the appliance. \n

**For the marked areas of image 2:** On the far right of the image, there's a notably round, possibly teardrop-shaped, vase. This vase seems to be the bulkiest among all, being referred to as the 'fat one'. It is situated in front of some objects which appear to be red. \n

**For the marked areas of image 3:** An arm wearing a purple sleeve is seen in the top right corner of the image, with a hand gripping a fork poised to cut into some food. \n

**For the marked areas of image 4:** A tall, lush potted plant can be observed amidst a setting of a white building. This plant, which could be described as a bush or large shrub, is positioned between green and red signs. It appears as a significant green element between these signs, contributing to the natural allure of the scene.

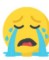

**Rejected: For the marked areas of image 1:** The right part of a stainless-steel oven door is visible in the corner of the image. It almost seems cut-off from the frame and includes a sturdy handle, only partially seen due to positioning. \n

**For the marked areas of image 2:** At the center of the image, there's a blue vase, distinctively sandwiched between two other vases. Its location in the very center of the image makes it a reference point, much like '12 noon' on a clock. \n

**For the marked areas of image 3:** A woman in a grey shirt is seen at the top of the image with her left hand gracefully placed on a sandwich. She garners further attention due to the shiny ring on her finger, indicating her fidelity to her husband. \n

**For the marked areas of image 4:** On the bottom left, there's a noticeably yellowish-green bush, planted in a pot. It's positioned interestingly, partially covering a blue sign and situated next to a sign indicating 'Moulin Hotel'. The position and color of this bush add an element of intrigue and a splash of green to the urban scene.

Figure 9: Example of Needle-Level preference pair with bbox region mismatches perturbation (Detail Misinterpret).

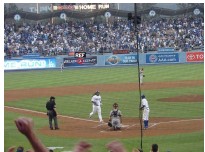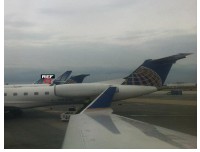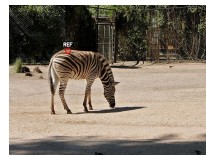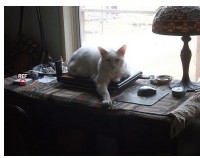

**Question:** <image><image><image><image>**Please provide a detailed description of the marked point in each image. The marked points are indicated by a red hollow circle with a 'REF' label positioned above the circle.**

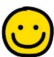

**Chosen: For the marked point of image 1:** The region shows a segment of a baseball field, focusing on the batter's area by home plate. Visible are players in uniforms engaged in the game, with one in stance to bat and a catcher and umpire in position behind him. The dirt-covered ground, home plate, and the players' equipment, like the batter's helmet, are prominent. \n

**For the marked point of image 2:** This region shows a commercial airplane tail with distinctive circular branding. The aircraft is stationary on the tarmac, and the tail appears prominently against a cloudy sky. \n

**For the marked point of image 3:** This is an image of a zebra positioned in profile, grazing on the ground. It is a full-grown adult with characteristic black and white striped fur. The mark is placed near the upper region of the zebra's body, close to the base of its neck. This particular stripe pattern is unique to each individual zebra, similar to a human fingerprint. The immediate area around the mark shows the zebra's mane, which consists of short, erect fur that follows along the neck's curve. \n

**For the marked point of image 4:** A notepad likely containing handwritten notes or drawings, surrounded by creative or writing utensils.

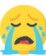

**Rejected: For the marked point of image 1:** This is a display of a large, horizontal electronic advertisement board located at the edge of the stands in a baseball stadium, showing a commercial message or sponsorship. The board appears to be illuminated and is part of the stadium's infrastructure, meant for the audience's viewing. \n

**For the marked point of image 2:** Focus is on the wingtip of an airplane, showing the curved end and navigation lights. The wing is against a backdrop of the sky ranging from pale to darker shades, possibly indicating cloudy weather.

**For the marked point of image 3:** This is an image of a zebra positioned in profile, grazing on the ground. It is a full-grown adult with characteristic black and white striped fur. The mark is placed near the upper region of the zebra's body, close to the base of its neck. This particular stripe pattern is unique to each individual zebra, similar to a human fingerprint. The immediate area around the mark shows the zebra's mane, which consists of short, erect fur that follows along the neck's curve. \n

**For the marked point of image 4:** A small, possibly decorative bowl that may be utilized for holding various small items or as a decor piece itself.

Figure 10: Example of Needle-Level preference pair with point region mismatches perturbation (Detail Misinterpret).

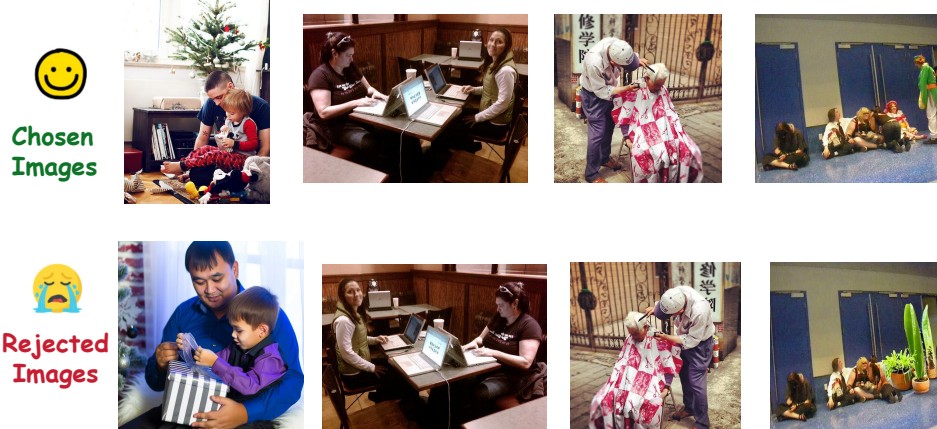

Question: <image><image><image><image>\n
**For image 1,** what can you tell me about the colors of the child's clothing while they're opening the gift, and can you describe the scene around them?**\n For image 2,** how is the thinner woman positioned in relation to the person in the black shirt in the image?**\n For image 3,** what can you tell me about the position of the elderly Asian man with white hair in relation to the Chinese characters in the image? Is he on the left or the right?**\n For image 4,** how many people are there in the image, and how many costumes can you see?

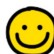

**Chosen: For image 1,** The child is dressed in red and gray while opening a gray and white striped gift, sitting on a man's lap who is wearing a blue shirt, and together they are positioned in front of a window and a small Christmas tree.\n

**For image 2,** The thinner woman is positioned to the right of the person in the black shirt.\n

**For image 3,** The elderly Asian man with white hair is to the right of the Chinese characters.\n

**For image 4,** There are 5 people and 6 costumes.

Figure 11: Example of Needle-Level preference pair with image contrastive perturbation (Detail Misinterpret).

