# OpenReview forum: "Zooming from Context to Cue: Hierarchical Preference Optimization for Multi-Image MLLMs"
_NeurIPS.cc/2025/Conference — NeurIPS 2025 poster_

### Official Review · Reviewer_iyTW · 2025-06-29

**Clarity:** 2
**Significance:** 3
**Originality:** 2
**Rating:** 4
**Confidence:** 3

**Summary:**

This paper aims to address common hallucinations in MLLMs when processing multi-image inputs, specifically context omission, conflation, and detail misinterpretation. To this end, the authors propose a hierarchical preference optimization framework named CCDPO. This framework is divided into two levels: the Context-Level, which optimizes the model's understanding of the global sequence through perturbation strategies like sequence truncation and content swapping , and the Needle-Level, which enhances the model's perception of local, key details through a combination of visual prompts and vision-contrastive learning. Additionally, the authors constructed an automatically generated preference dataset, MultiScope-42k, to support the training of this framework. Experimental results indicate that this method achieves performance gains across multiple multi-image and single-image benchmarks.

**Questions:**

See weaknesses.

**Ethical Concerns:**

["NO or VERY MINOR ethics concerns only"]

**Final Justification:**

The authors have solved most of my concerns.

**Limitations:**

yes

**Quality:**

3

**Strengths And Weaknesses:**

Strengths:
1. The paper provides a valuable and clear categorization of hallucinations in multi-image understanding, identifying them as context omission, context conflation, and detail misinterpretation. This classification offers distinct targets and a focused direction for the methodological design that follows.
2. The model is evaluated extensively on a wide range of multi-image and single-image benchmarks, with performance compared against relevant baselines like SFT and MIA-DPO. Additionally, the paper includes detailed ablation studies to validate the effectiveness of the framework's different components.

Weaknesses:
1. The paper's core contribution lies in its novel combination of existing techniques rather than a fundamental methodological breakthrough. DPO and its applications in multimodal contexts are previously established, with the paper itself citing MIA-DPO as a baseline for multi-image tasks. The vision-contrastive learning at the needle-level also explicitly draws from prior work. Therefore, the work presents itself more as a sophisticated engineering and integration effort, with its methodological novelty being somewhat limited.


2. The ablation study does not analyze the individual effects of the Needle-Level and Context-Level optimizations. A more direct comparison could have provided better insight into the contribution of each method to downstream task performance.

3. In the ablation study tables, shorthand terms like Needle-Levi-TDPO and Needle-Levi-VDPO are used without being clearly defined in the main text of the paper. This lack of explicit definition can cause confusion for the reader.

---

> ### Author Rebuttal · Authors · 2025-07-31
>
> > ### **W1**: The framework's novelty lies in its hierarchical application and combination of existing mechanisms, rather than in the creation of new fundamental components.
>
> 1. Firstly, let us outline the differences between our proposed approach, CcDPO, and MIA-DPO:
> - One key difference between CcDPO and MIA-DPO lies in the design of **question prompts**.
>   - **MIA-DPO solely associate each question with a specific image, which causes MIA-DPO to only consider a subset**  (instead of the entire set) of images to infer the final answer. This design limits the model's effectiveness when handling questions that require information from all images to arrive at the correct answer.
>   - CcDPO uses multi-image captioning prompts (two-stage: coarse-grained context → fine-grained "needle" analysis) to enhance **full-image context understanding** with minimal construction cost (Table 1).
>   - In addition, visual prompts in the needle stage **force the model to rely on the marked information within the images**, enhancing its ability to interpret fine details accurately while reducing the likelihood of hallucinations.
>
> - Another key difference between CcDPO and MIA-DPO is the **construction cost** of preference pairs.
>   - MIA-DPO requires pre-filtering rejected responses based on base model predictions, which introduces computational overhead and noise due to potential discrepancies between attention values and final answers.
>   - CcDPO leverages **structured captioning for efficient negative sample generation (via swapping/truncating captions)**, simulating key multi-image issues (Context Omission, Conflation, Detail Misinterpretation) at low cost.
> 2. Next, let us outline the differences between our CcDPO and prior vision-contrastive work  [49, 32]:
>   - Prior methods align global image semantics but **miss critical instance-level details**; CcDPO uses visual prompts to **anchor regions/objects**, boosting fine-grained tasks (Counting, Attribute Understanding).
>   - While prior vision-contrastive learning methods primarily focus on single-image tasks, our work first extends this concept into the **multi-image understanding domain**.
>
> 3. Finally, let's summarize the contributions of the paper:
> - We created a caption generation task to evaluate MLLMs' multi-image understanding, **identifying a key limitation: the inability to integrate partial context** (Figure 2 and Table 2 of main paper), which hinders complex reasoning. We also pinpoint three hallucination types: **Context Omission, Context Conflation, and Detail Misinterpretation.**
> - To address this, we propose a cost-effective two-level preference optimization framework that improves multi-image perception by leveraging sequential context and fine-grained details, **achieving up to a +4.6 point performance boost** (Table 1 in our main paper).
> - We also introduce MultiScope-42k, a large-scale, auto-generated DPO dataset that **structures image annotations into sequences** with rich insights. This reduces data collection costs and, using a coarse-to-fine pairing strategy, delivers better generalization than SFT on multi-image tasks.  (refer to the responses of Reviewer ut67.)
>
> > ### **W2**: The ablation study does not analyze the individual effects of the Needle-Level and Context-Level optimizations.
>
> Thank you for your valuable feedback. To address your concerns, we conducted additional analyses to evaluate the individual effects of Needle-Level and Context-Level optimizations.
> - The benchmark results below demonstrate that both optimizations independently improve performance, with Needle-Level excelling at fine-grained detail-oriented tasks (e.g., **MuirBench +3.8%**), and Context-Level performing better in holistic context-reasoning tasks (e.g., **Mantis +6.6%, Q-Bench2 +2.6%**). When combined, they achieve the best overall results, significantly outperforming the baseline (e.g., **MuirBench +6.1%, Mantis +9.2%, BLINK +4.8%, Q-Bench2 +3.8%**).
>
> | Context-Level | Needle-Level | **MuirBench** | **Mantis** | **BLINK** | **Q-Bench2** |
> |---------------|--------------|----------------|------------|-----------|--------------|
> | ×| ×| 42.5 | 60.4 | 51.1 | 73.8 |
> | ✔| × | 44.4 | 67.0 | 54.8 | 76.4 |
> | ×| ✔ | 46.3 | 65.0  | 54.3| 75.5 |
> | ✔| ✔ | **48.6** | **69.6** | **55.9** | **77.6** |
> - The table below provides additional insights into the effects on MuirBench, showing that Needle-Level optimization improves tasks requiring fine-grained perception, such as **Attribute Understanding (+14.4%) and Counting (+6.9%)**, while Context-Level optimization enhances tasks involving contextual reasoning, such as **Diagram Understanding (+2.5%) and Difference Spotting (+9.1%)**. These results highlight that the two optimizations are complementary, and their integration effectively resolves challenges in multi-image detail recognition and context reasoning, achieving state-of-the-art performance.
>
> | Model | Overall | Action Understanding | Attribute Understanding| Cartoon Understanding | Counting | Diagram Understanding | Difference Spotting | Geographic Understanding | Image-Text Matching | Ordering | Scene Understanding | Visual Grounding | Visual Retrieval |
> | :--- | :---: | :---: | :---: | :---: | :---: | :---: | :---: | :---: | :---: | :---: | :---: | :---: | :---: |
> | LlaVA-OV | 42.5 | 35.9 | 33.1 | **35.8** | 24.7 | 55.0 | 30.0 | **46.0** | 46.9 | **20.3** | 72.0 | 29.7 | 47.6 |
> | Needle-Level | 46.3 | **39.0** | **47.5** | 32.1 | **31.6** | 54.8 | 35.0 | 43.0 | **55.0** | 17.2 | **74.7** | **33.3** | **47.8** |
> | Context-Level | 44.4 | 37.8 | 39.8 | 33.3 | 29.5 | **57.5** | **39.1** | 39.0 | 50.4 | 14.1 | 64.0 | 27.4 | 44.2 |
>
> > ### **W3**: The shorthand terms like Needle-Levi-TDPO and Needle-Levi-VDPO are used without being clearly defined in the main text of the paper.
>
> Apologies for any confusion caused.
>
> **Needle-Level Language-based Preference Optimization (Needle-Levl-TDPO)** focuses on refining the model's preferences based on textual information. It uses paired textual descriptions ($y_{chosen}, y_{rejected}$) to guide the model's attention to specific regions within an image ($x$). The optimization process ensures that the model learns to associate descriptions with the highlighted regions, favoring preferred regions as follows:
> $$ (x, y_{\text{chosen}}) \succ (x, y_{\text{rejected}}) $$
>
> **Needle-Level Vision Contrastive Preference Optimization (Needle-Levl-VDPO)** forces the model to make preference judgments based on visual information. It works by constructing pairs of images ($x_{chosen}, x_{rejected}$) with preferences controlled as variables, guiding the model to distinguish and prioritize chosen visual regions. This process is expressed as:
> $$ (x_{\text{chosen}}, y) \succ (x_{\text{rejected}}, y) $$
>
> We will ensure these terms are clearly and thoroughly defined in the main text of the revised version to improve accessibility and understanding for all readers, including non-technical audiences.

---

> > ### Comment · Reviewer_iyTW · 2025-08-05
> > **Response to rebuttal**
> >
> > Thank you for the detailed response.
> > I think the authors have solved most of my concerns.
> > I will raise my score to borderline accept.

---

> > > ### Author Response · Authors · 2025-08-06
> > > **Thanks!**
> > >
> > > Thank you very much for raising the score! We appreciate your constructive review. Your insights into the ablation study of the individual effects of the Needle-Level and Context-Level DPO are crucial for enhancing the robustness of our work. Additionally, your guidance on clarifications, such as defining shorthand terms clearly, has been extremely helpful and significantly improved our paper’s readability. Thank you again for your time and valuable contributions!

---

### Official Review · Reviewer_wac8 · 2025-07-01

**Clarity:** 4
**Significance:** 4
**Originality:** 3
**Rating:** 5
**Confidence:** 3

**Summary:**

This paper proposes Context-to-Cue Direct Preference Optimization (CcDPO), a hierarchical framework to improve multi-image understanding in Multi-modal Large Language Models (MLLMs). The method tackles three common hallucination issues - context omission, context conflation, and detail misinterpretation - by introducing two levels of preference alignment: (1) Context-Level Optimization, which promotes accurate and complete multi-image comprehension through structured per-image captions and perturbation-based training; and (2) Needle-Level Optimization, which enhances fine-grained perception using region-level visual prompts and contrastive supervision. To support training, the authors construct MultiScope-42k, a large-scale dataset with automatically generated multi-level preference pairs. Experiments on a wide range of multi-image and single-image benchmarks demonstrate that CcDPO consistently improves performance and reduces hallucination compared to prior methods like MIA-DPO.

**Questions:**

The paper presents an impressive contribution with the CcDPO framework, supported by a robust dataset (MultiScope-42k) and thorough evaluations. My only minor concern is in Table 1, where the bolded result for Qwen2-VL on the MIRB task does not correspond to the highest score, which may confuse readers. Please clarify if this is intentional or correct the bolding, and consider verifying other tables for consistency. This small revision would further polish an already strong submission.

**Ethical Concerns:**

["NO or VERY MINOR ethics concerns only"]

**Final Justification:**

The authors have addressed all of my comments. In addition to correcting the formatting error in Table 1, they provided a detailed comparison between CcDPO and prior approaches, clarified its practical advantages, and extended their experiments to the video domain, demonstrating generalizability. These updates reinforce the paper’s originality and impact. I maintain my rating of 5.

**Limitations:**

Yes

**Paper Formatting Concerns:**

My only minor concern is a formatting issue in Table 1 as I mentioned in Questions, where the bolded result for Qwen2-VL on the MIRB task does not highlight the highest score, which may cause confusion.

**Quality:**

4

**Strengths And Weaknesses:**

Strengths:
The paper presents a well-motivated and clearly described approach to improving multi-image understanding in MLLMs, addressing common hallucination issues. The proposed CcDPO framework is conceptually sound, combining structured context-level and fine-grained needle-level preference optimization. The methodology is supported by a newly constructed dataset, MultiScope-42k, which enables scalable training. The empirical results are thorough, covering a wide range of benchmarks, and show consistent improvements over strong baselines. The writing is clear and the paper is easy to follow.

Weaknesses:
The overall design is solid, but the method mainly combines existing techniques such as DPO and visual prompting in a hierarchical way, rather than introducing fundamentally new mechanisms. In addition, as the authors acknowledge, the framework does not explicitly handle temporal reasoning, which may limit its applicability to video-related tasks.

---

> ### Author Rebuttal · Authors · 2025-07-31
>
> > ### **W1(1)**: The framework's novelty lies in its hierarchical application and combination of existing mechanisms, which results in somewhat limited contributions.
>
> 1. Firstly, let us outline the differences between our proposed approach, CcDPO, and MIA-DPO:
> - One key difference between CcDPO and MIA-DPO lies in the design of **question prompts**.
>   - **MIA-DPO solely associate each question with a specific image, which causes MIA-DPO to only consider a subset**  (instead of the entire set) of images to infer the final answer. This design limits the model's effectiveness when handling questions that require information from all images to arrive at the correct answer.
>   - CcDPO uses multi-image captioning prompts (two-stage: coarse-grained context → fine-grained "needle" analysis) to enhance **full-image context understanding** with minimal construction cost (Table 1).
>   - In addition, visual prompts in the needle stage **force the model to rely on the marked information within the images**, enhancing its ability to interpret fine details accurately while reducing the likelihood of hallucinations.
>
> - Another key difference between CcDPO and MIA-DPO is the **construction cost** of preference pairs.
>   - MIA-DPO requires pre-filtering rejected responses based on base model predictions, which introduces computational overhead and noise due to potential discrepancies between attention values and final answers.
>   - CcDPO leverages **structured captioning for efficient negative sample generation (via swapping/truncating captions)**, simulating key multi-image issues (Context Omission, Conflation, Detail Misinterpretation) at low cost.
> 2. Next, let us outline the differences between our CcDPO and prior vision-contrastive work  [49, 32]:
>   - Prior methods align global image semantics but **miss critical instance-level details**; CcDPO uses visual prompts to **anchor regions/objects**, boosting fine-grained tasks (Counting, Attribute Understanding).
>   - While prior vision-contrastive learning methods primarily focus on single-image tasks, our work first extends this concept into the **multi-image understanding domain**.
>
> 3. Finally, let's summarize the contributions of the paper:
> - We created a caption generation task to evaluate MLLMs' multi-image understanding, **identifying a key limitation: the inability to integrate partial context** (Figure 2 and Table 2 of main paper), which hinders complex reasoning. We also pinpoint three hallucination types: **Context Omission, Context Conflation, and Detail Misinterpretation.**
> - To address this, we propose a cost-effective two-level preference optimization framework that improves multi-image perception by leveraging sequential context and fine-grained details, **achieving up to a +4.6 point performance boost** (Table 1 in our main paper).
> - We also introduce MultiScope-42k, a large-scale, auto-generated DPO dataset that **structures image annotations into sequences** with rich insights. This reduces data collection costs and, using a coarse-to-fine pairing strategy, delivers better generalization than SFT on multi-image tasks.  (refer to the responses of Reviewer ut67.)
>
> > ### **W1(2)**: The applicability of the proposed framework to video-related tasks.
>
> Thank you for your valuable feedback. As acknowledged in the limitations section, we did not specifically construct preference pairs for video data. However, our approach can be easily extended to video data using a structured format, such as **<video clip 1> <caption 1> <video clip 2> <caption 2>**, while incorporating the proposed Sequence Truncation and Content Swapping strategies for generating negative samples. This allows video DPO datasets to be created at minimal cost.
>
> To further support this, we constructed a 7k video dataset based on the described methodology and performed DPO training. The table below illustrates the effectiveness of our framework in VideoMME tasks:
>
> | Model         | Overall | Temporal Perception | Spatial Perception | Attribute Perception | Action Recognition | Object Recognition | OCR Problems | Counting Problem | Temporal Reasoning | Spatial Reasoning | Action Reasoning | Object Reasoning | Information Synopsis |
> |:-------------:|:-------:|:-------------------:|:------------------:|:--------------------:|:------------------:|:------------------:|:------------:|:----------------:|:------------------:|:-----------------:|:----------------:|:----------------:|:---------------------:|
> | LlaVA-OV      | 53.7    | 0.491               | 0.556              | 0.685               | 0.505              | 0.559              | **0.568**        | 0.354            | 0.384              | 0.714             | **0.523**            | **0.518**            | **0.675**                  |
> | +CcDPO-Video          | **54.4**    | **0.509**               | **0.574**              | **0.698**               | **0.527**              | **0.582**              | 0.547        | **0.377**            | **0.401**              | **0.732**             | 0.519            | 0.511            | 0.663                  |
>
> It is worth noting that even an improvement of 1 point in video understanding tasks is significant—for instance, prior video DPO work [1,2] achieves improvements in the range of 0.4–1.2 points. While our CcDPO method is not specifically tailored for video understanding, **its use of sequence descriptions as a proxy shows potential in enhancing sequence comprehension.** Moreover, the proposed negative sample **perturbation strategies are generalizable**, making the construction process more cost-effective by reducing reliance on model re-labeling or rejection sampling strategies.
>
> ---
> References
>
> [1] VideoPASTA : 7K Preference Pairs That Matter for Video-LLM Alignment
>
> [2] Direct preference optimization of video large multimodal models from language model reward.
>
> > ### **Q1**: Bold errors of Table 1.
>
> Apologies for my mistake. We have corrected this error and have also verified the other tables to ensure consistency.

---

> > ### Comment · Reviewer_wac8 · 2025-08-02
> > **Acknowledgement of rebuttal**
> >
> > Thank you for the detailed rebuttal and the clarifications.
> >
> > I appreciate that you corrected the bolding error in Table 1 and verified the other tables for consistency. I have no further questions and will take your responses into account when finalizing my review.

---

> > > ### Author Response · Authors · 2025-08-04
> > > **Thanks!**
> > >
> > > We sincerely thank the reviewer for acknowledging our efforts and providing valuable feedback, which helped us identify areas of ambiguity in our paper. Thank you again for your time and effort in reviewing this paper!

---

### Official Review · Reviewer_ut67 · 2025-07-02

**Clarity:** 2
**Significance:** 1
**Originality:** 1
**Rating:** 2
**Confidence:** 4

**Summary:**

This paper introduces CcDPO (Context-to-Cue Direct Preference Optimization) as a hierarchical preference optimization framework for improving multi-image understanding problems of Multimodal Large Language Models (MLLMs). The authors identify three prevalent types of hallucinations—context omission, context conflation, and detail misinterpretation—and propose a two-level optimization strategy to address them: (i) Context-Level Optimization improves holistic sequence comprehension via structured per-image captions and perturbation-based contrastive training; (ii) Needle-Level Optimization enhances fine-grained perception using region-level visual prompts and vision-language contrastive objectives. The proposed method is trained on a newly constructed dataset, MultiScope-42k, and achieves improvements across multiple multi-image benchmarks (e.g., MUIRBench, MIBench, MIRB).

However, the authors do not include any ablation or comparative experiments where CcDPO is trained using existing public datasets (e.g., LLaVA-23K, MDVP, MVC) to isolate the contribution of the optimization method itself versus the data quality and scale. Also, I do not find any supplementary materials or documentation that would help reproduce the reported results using the released code. The provided README appears to be identical to that of LLaVA-NeXT: Open Large Multimodal Models, without any specific instructions for training or evaluating CcDPO.

**Questions:**

1. Could the authors clarify whether the released codebase includes any implementation of CcDPO beyond what is already available in LLaVA-NeXT? A readme of CcDPO is preferred in response.

2. Could the authors provide an ablation study where CcDPO is trained without MultiScope-42k, for example using only existing datasets such as LLaVA-23K or MDVP? Also, the baselines trained with MultiScope-42k is also necessary for verification.

3. The two optimization components—context-level and needle-level—appear to be trained independently and applied in parallel. Could the authors clarify whether there is any interaction or dependency between them during training or inference?

The major limitations are stated in the weakness part.
I encourage the authors to respond the ablation study of MultiScope-42k, as well as the readme of codes for reproduction.

**Ethical Concerns:**

["NO or VERY MINOR ethics concerns only"]

**Final Justification:**

I recommend **rejecting this paper**. From the initial submission through the rebuttal, the authors have consistently attempted to blur the line between the performance gains contributed by their proposed algorithm (CcDPO) and those arising from the introduction of a new dataset (MultiScope-42k).

Figure 1 is a particularly **misleading visualization**. At first glance, it appears to suggest that CcDPO yields significant performance improvements. However, upon closer examination of the provided tables, it becomes clear that the majority of the gains are attributable to the dataset rather than the algorithm itself.

Moreover, the authors did not provide a proper README for their code, further hindering reproducibility and transparency. Taken together, these issues point to a systematic attempt to obscure the true source of improvements, which raises concerns about the credibility of the empirical findings.

Lastly, **rejecting this paper is necessary to uphold the value of genuine algorithmic innovation**. If the community starts to accept papers that primarily introduce new datasets but mask them as algorithmic contributions, it sets a dangerous precedent. It risks favoring researchers with the resources to construct larger datasets, while **penalizing those focused on advancing algorithm design — even if their improvements are more principled but less dramatic in performance**. This trend would ultimately undermine progress in core methodological research.

**Limitations:**

Yes

**Quality:**

2

**Strengths And Weaknesses:**

Pros:
1. The paper addresses a critical challenge in multi-image understanding. The authors clearly articulate the importance of this issue and provide a thoughtful taxonomy of error types, which helps ground the methodological design.

2. The authors present a carefully constructed dataset (MultiScope-42k) tailored for hierarchical preference optimization, which effectively supports both context-level and fine-grained training objectives.

3. The proposed two-level optimization framework (CcDPO) is methodologically sound and conceptually intuitive.

Cons:
1. The released code lacks any specific README or documentation related to CcDPO. It appears to be identical to the LLaVA-NeXT repository, with no visible modifications or guidance for reproducing the results. This makes the authors’ claim of code release less convincing and significantly limits reproducibility.

2. There is no ablation study evaluating CcDPO without the proposed MultiScope-42k dataset. As a result, it is unclear whether the performance gains stem from the optimization method itself or simply from the quality and scale of the new dataset. This is unfair for the baseline models that cannot access the MultiScope-42k datasets.

3. The two proposed optimization stages—context-level and needle-level—are designed independently and operate in parallel. There is no clear logical or learning dependency between them, which weakens the hierarchical narrative implied in the method’s name and motivation.

---

> ### Author Rebuttal · Authors · 2025-07-31
>
> > ### **W1,Q1(1)**: Does the released code include any new CcDPO implementations not already available in LLaVA-NeXT?
>
> Indeed, we made the following modifications:
> 1. **Data Processing**: Added `llava/train/train_vdpo.py` and `llava/train/train_svco_triples.py` to process inputs containing both text- and image-level positive and negative samples.
> 2. **Model Training**: Added `trl/trainer/vdpo_trainer.py` and `trl/trainer/vdpo_trainer_svco_triples.py` to implement the Hybrid Visual-Language Optimization formulas (1), (2), and (3) as described in our paper.
>
> > ### **W1,Q1(2)**: The provided code lacks a README for CcDPO.
>
> We apologize for the initial lack of documentation due to the submission deadline. The code is accurate, and we have now provided a detailed guide below to help reproduce results.
> #### **Data**
> - Training Data: In this work, we use the constructed MultiScope Dataset with 42k training samples as our training dataset.​
> - Evaluation Data: The evaluation data can be downloaded from project homepage. The evaluation datasets utilized in our work are listed below: MuirBench; MIRB; BLINK; Mantis; NLVR2; MIBench; Q-Bench2
>
> #### **Install**
>
> 1. Clone this repository and navigate to the source folder:
> > cd LLaVA-NeXT
> 2. Build Environment (key package versions: transformers=4.45.2，accelerate=1.4.0, deepspeed=0.15.4，peft=0.14.0)
> > pip install -e .
>
> #### **LLaVA-OV CcDPO Training**
> Configure the training dataset ”DATA_PATH“ and image folder "image_folder" and the checkpoint name “OUTPUT_DIR“.
> - Stage1: set ”DATA_PATH“ to Context-Level DPO dataset.
> > bash scripts/train/vdpo_ov7b.sh
> - Stage2: set ”DATA_PATH“ to Needle-Level Language-based DPO dataset.
> > bash scripts/train/vdpo_ov7b.sh
> - Stage3: set ”DATA_PATH“ to Needle-Level Vision-based DPO dataset.
> > bash scripts/train/svco_ov7b.sh
>
> #### **Evaluation**
> The trained models are evaluated on MuirBench, BLINK and MIBench benchmarks using VLMEvalKit tools.
> > cd VLMEvalKit
> >
> > CUDA_VISIBLE_DEVICES=0,1,2,3,4,5,6,7 torchrun --nproc-per-node=8 --master_port 3951 run.py --data BLINK --model llava_onevision_qwen2_7b_ov --verbose --work-dir xxx
>
> The trained models are evaluated on Mantis, NLVR2, and Q-Bench2 benchmarks using Mantis's code.
>
> > cd Mantis-main/mantis/benchmark
> >
> > CUDA_VISIBLE_DEVICES=0 torchrun --nproc-per-node=1 --master_port 1921 eval.py --dataset_path "mantis_eval" --dataset_name "mantis_eval" --model_name llava-ov --results_dir xxxx --overwrite True
>
> ---
>
> > ### **W2,Q2**: There is no ablation study to disentangle the performance gains from the CcDPO method versus the new MultiScope-42k dataset.
>
> Thank you for your insightful comments. To address your concern, we conducted three sets of experiments to verify **Method Effectiveness**, **Dataset Effectiveness**, and **Dataset Scale Evaluation**, with the following conclusions:
> - We trained CcDPO using existing public single-image or multi-image input DPO datasets, and **our CcDPO still achieved better performance compared to the baseline** SFT method. (Table 1)
> - We compared the effectiveness of DPO across different datasets and found that **single-image QA with multi-image inputs is not optimal** for addressing multi-image hallucinations—multi-image QA works better. In this scenario, **captioning is the lowest-cost method**. (Table 1)
> - When matched in size with the compared datasets, our MultiScope-42k still performs better, and **performance improves as the size increases.** (Table 2)
>
> We will now analyze each issue in detail:
> > ### **W2,Q2(1)**: Method Effectiveness：there is no ablation study evaluating CcDPO without the proposed MultiScope-42k dataset  (e.g., LLaVA-23K, MDVP).
>
> To clarify, the suggested LLaVA-23k and MDVP datasets are primarily SFT datasets and are not directly applicable to DPO training. Therefore, we selected two existing DPO datasets to train CcDPO, including the multi-image input single-image QA dataset MIA-DPO, the **single-image input single-image QA** dataset POVID 17k[1] + RLHF-V 6k[2], and our **multi-image input multi-image QA** dataset MultiScope. We compared CcDPO with the baseline SFT method, as shown in the table below:
>
> | Description of Data | Method|Training Dataset| MuirBench | Mantis | BLINK | Q-Bench2 |
> |-|-|-|--|--|--|-|
> |    | LLaVA-OV | -   | 42.5  | 60.4   | 51.1  | 73.8     |
> |**MIA-DPO-28k**: multi-images input, single image QA| LLaVA-OV + SFT  | MIA-DPO-28k | 42.6  | 59.0   | 52.3  | 74.0     |
> | | LLaVA-OV + CcDPO | MIA-DPO-28k | **42.9**  | **62.7**| **53.2** | **74.8** |
> | **POVID-17k** + **RLHF-V-6k**: single-image input, single image QA | LLaVA-OV + SFT     | POVID-17k + RLHF-V-6k         | 43.3      | 59.4   | 51.7  | 73.5     |
> |    | LLaVA-OV + CcDPO| POVID-17k + RLHF-V-6k | **44.7**  | **61.8**| **54.1** | **75.6** |
> | **MultiScope**: multi-images input, multi-image QA | LLaVA-OV + SFT| MultiScope-42k| 45.4      | 64.9   | 53.4  | 75.7     |
> | | LLaVA-OV + CcDPO        | MultiScope-42k  | **48.6**  | **69.6**| **55.9** | **77.6** |
>
> Across all datasets, CcDPO consistently outperforms the baseline method, showcasing its ability to leverage contrastive information from preference data. Its effectiveness in multi-image tasks, however, is limited with MIA-DPO-28k or POVID-17k + RLHF-V-6k—**likely because these datasets lack tasks explicitly involving multiple images in final answers.** In contrast, the low-cost, automatically constructed MultiScope dataset includes multi-image tasks that better utilize CcDPO’s capabilities, yielding its best performance.
>
> [1] RLHF-V: Towards Trustworthy MLLMs via Behavior Alignment from Fine-grained Correctional Human Feedback
>
> [2] Aligning modalities in vision large language models via preference fine-tuning.
> > ### **W2,Q2(2)**: Dataset Effectiveness: Do performance gains stem solely from the new dataset’s quality?
>
> - To clarify, **performance gains stem from both the data and the CcDPO method**. As the ablove table shows, while MultiScope outperforms other public datasets for multi-image tasks, integrating our CcDPO method yields a maximum average performance **improvement of ~3.1%**. This is mainly attributed to our core idea—**using multi-image descriptions as an alternative approach to understanding context**.
> Our dual-level optimization forces the model to describe each image accurately from coarse to fine, cutting multi-image hallucinations by up to **26%** (Table 2 of main paper) and boosting its ability to grasp multi-image context.
> - In contrast, CcDPO cannot fully exert its effect with other datasets, as their QA tasks only require leveraging local information within image sequences.
>
> > ### **W2,Q2(3)**:  Do performance gains stem solely from the new dataset’s scale?
>
> We analyze the impact of training size by performing an ablation study using the same total number of preference pairs as MIA-DPO **(more data filtering details and analysis are in Supplementary Section C)**. The results are shown below.
>
> | Dataset |Size|MuirBench|MIRB| BLINK | Mantis | NLVR2 | MIBench | Q-Bench2 | Average |
> |--|-|--|--|--|---|--|--|--|--|
> | LLaVA-OV| - | 42.5| 47.3  | 51.1  | 60.4   | 89.4  | 73.6 | 73.8   | 62.5    |
> | + MIA-DPO | 28.9K  |41.4| 48.0  | 53.7  | 60.3   | 88.2  | 67.8    | 74.0     | 61.9    |
> | + CcDPO | 28.1K | 46.7 | 51.2| **56.5** | 69.1 | **90.7** | 72.1 | **79.3** | 66.5 |
> | + CcDPO | 41.8K| **48.6** | **51.4** | 55.9 | **69.6** | **91.2** | **75.2** | 77.6 | **67.1** |
> | Δ   | - | +6.1 | +4.1  | +4.8  | +9.2   | +1.8  | +1.6    | +3.8     | +4.6    |
>
> Results show that, at the matched training size, our CcDPO consistently outperforms MIA-DPO across benchmarks, highlighting the effectiveness of its structured dual-level supervision. Training with the full preference set delivers the best average performance, demonstrating the benefits of high-quality, large-scale supervision. This also reveals a trade-off between contextual alignment and perceptual precision.
>
> ---
>
> > ### **W3,Q3**: The two proposed optimization stages—context-level and needle-level—are designed independently and operate in parallel.
>
> Thank you for your thoughtful comments. We would like to clarify that the two optimization stages—context-level and needle-level DPO—are **sequentially designed** and **mutually dependent**, reflecting a coarse-to-fine cognitive process. Specifically:
> - **Sequential Design:** The model trained with context-level DPO serves as **the initialization** for needle-level DPO training, establishing a **sequential workflow.**
>   - In the context-level stage, the model learns to integrate global information, addressing foundational errors like **context omission or conflation.**
>   - Following this, needle-level DPO focuses on **refining detail misinterpretation** by enhancing perceptual precision for critical regions. This workflow mirrors human general reasoning for multi-image problems: one first understands the global context, then zooms in to focus on specific details as needed for the task.
> - **Stage Dependency:** To explore interactions and dependencies between context-level and needle-level DPO, we conducted ablation experiments with varying training stage orders and mixed training approaches. Results are provided below：
>
> | Configuration | MuirBench | Mantis | BLINK | Q-Bench2 |
> | :- | :-: | :-: | :-: | :-: |
> | Baseline | 42.5 | 60.4 | 51.1 | 73.8 |
> | One-Stage Training | 46.2 | 65.4 | 55.1 | - |
> | Context-Level Only | 44.4 | 69.0 | 55.3 | 76.4 |
> | Needle-Level Only | 46.3 | 65.0 | 54.3 | 75.5 |
> | Needle-Level -> Context-Level | 46.5 | 66.3 | 54.1 | **77.8** |
> | Context-Level -> Needle-Level | **48.6** | **69.6** | **55.9** | 77.6 |
>
> Results demonstrate that training context-level DPO before needle-level DPO consistently yields optimal performance across benchmarks, emphasizing a hierarchical and interdependent relationship. Skills learned in the context-level stage (e.g., coherent global reasoning) establish a strong foundation for the needle-level stage to achieve fine-grained perceptual precision.

---

> > ### Comment · Reviewer_ut67 · 2025-08-06
> >
> > Thank the authors for the response.
> >
> > 1. Thank you for your readme. Simply appending the codes without any documentation does no help to validate the reproducibility
> >
> > 2. From Table in W2,Q2(1), it is evident that the primary performance gain comes from the newly constructed MultiScope-42k dataset, rather than the CcDPO method itself. When CcDPO is trained on public datasets such as MIA-DPO or POVID + RLHF-V, the improvement over baselines is relatively modest, suggesting that the performance boost is largely attributed to the data rather than the optimization technique.
> >
> > More critically, the table lacks a key baseline: MIA-DPO trained on the MultiScope-42k dataset. The authors use “LLaVA-OV + SFT” as a baseline, but SFT is not an appropriate baseline for DPO-style training. Without a direct comparison between MIA-DPO and CcDPO on the same dataset (i.e., MultiScope), it is difficult to isolate the effect of the proposed method.
> >
> > Furthermore, MIA-DPO has been shown in its original paper to outperform SFT, yet in the authors’ Table 1, LLaVA-OV + MIA-DPO underperforms compared to SFT. This discrepancy raises concerns about whether the reproduction of MIA-DPO was successful or faithful to the original implementation.
> >
> > Overall, while the paper strongly emphasizes the effectiveness of the proposed CcDPO method, it downplays the decisive role of the newly introduced MultiScope-42k dataset. The lack of a fair comparison with the only relevant baseline MIA-DPO trained on the same dataset makes it difficult to isolate and verify the true contribution of CcDPO itself.

---

> ### Author Response · Authors · 2025-08-07
> **Round 1**
>
> > Q1：About Reproducibility.
>
> We sincerely apologize again for the oversights in the initial submission.
> - **Due to NIPS rebuttal policies prohibiting external links**, we cannot directly re-provide the updated code repository.
> However, we commit to promptly releasing full, well-documented source code with training scripts and model weights in a public repository upon manuscript acceptance, ensuring reproducibility.
> - To further assist, we suggest that reviewers can conduct ablative reproductions of the key components of our method using publicly available DPO datasets, **following the guidance in the readme document provided in our W1 response.**  This will allow for direct verification of the core mechanisms of our approach.
>
> ----------------------------
> > Q3: Disentangling Contributions of CcDPO itself– addressing the lack of MIA-DPO trained on MultiScope-42k.
>
> Thank you for your valuable feedback. Our baseline (MIA-DPO, a one-stage Vanilla DPO method) trained on MultiScope-42k has already been **provided in Row 2 of the W3Q3 table**. As indicated in the table below, CcDPO **outperformed MIA-DPO by a maximum of 4.2 points**—this clear improvement strongly validates the effectiveness of CcDPO's two-level DPO strategy. Additionally, we derived two further conclusions:
> - **The CcDPO method itself is demonstrably effective.**  Comparing the 4th row to the last row, CcDPO’s coarse-to-fine optimization order is a notable contribution. Without this order (4th row), improvements over Vanilla DPO are suboptimal. This confirms our performance **gains derive not only from the dataset but also from our tailored two-stage coarse-to-fine DPO approach.**
> - **The synergy between our two-level DPO strategy and MultiScope-42k is critical.** Compared to using either component alone, their combination yielded a maximum 9.2-point improvement on Mantis. This is because CcDPO requires accurate preference optimization for both full contexts and local details, while MultiScope-42k provides context-details positive/negative samples to enable this coordination. **Decoupling reduces effectiveness**, explaining relatively smaller gains on **other datasets (which lack samples covering both global and local info)**.
>
> | Method                          | Dataset         | MuirBench | Mantis | BLINK | Q-Bench2 |
> |---------------------------------|-----------------|-----------|--------|-------|----------|
> | LLaVA-OV                        | -  | 42.5      | 60.4   | 51.1  | 73.8     |
> | SFT                             | MultiScope-42k  | 45.4      | 64.9   | 53.4  | 75.7     |
> | MIA-DPO (One-Stage Vanilla DPO) | MultiScope-42k  | 46.2      | 65.4   | 55.1  | 76.1        |
> | Needle-Level -> Context-Level | MultiScope-42k  | 46.5 | 66.3 | 54.1 | **77.8** |
> | Context-Level -> Needle-Level (Ours) | MultiScope-42k  | **48.6** | **69.6** | **55.9** | 77.6 |
>
> ----------------------------
> > Q4: Addressing the misunderstanding that "LLaVA-OV + MIA-DPO underperforms compared to SFT"
>
> Thank you for pointing out this observation—we appreciate the opportunity to clarify this point, as it helps us address an ambiguity in our presentation.
>
> 1.**The claim that "LLaVA-OV + MIA-DPO underperforms compared to SFT" may stem from a misunderstanding.** To clarify, the SFT results in Table 1 of the main paper use our MultiScope-42k dataset, not the MIA-DPO dataset. This has made us aware of the ambiguity in Table 1, and we will explicitly specify the training dataset for each method in the revised version to improve readability.
>
> 2.Instead, **our conclusion aligns with MIA-DPO: DPO outperforms direct SFT.** This is verified by comparing LLaVA-OV + MIA-DPO (Table 1 of the main paper) with LLaVA-OV + SFT (trained on MIA-DPO-28k, from our response to W2,Q2(1)). As shown below, MIA-DPO outperforms SFT by **1.3 points on BLINK and 1.4 on Mantis**—consistent with **MIA-DPO’s Table 11 (average 1-point improvement)**, further validating the credibility of our results.
>
> | Method               | Training Dataset | Mantis | BLINK | Avg.  |
> |----------------------|------------------|--------|-------|-------|
> | LLaVA-OV + SFT       | MIA-DPO-28k      | 59.0   | 52.3  | 55.6 |
> | LLaVA-OV + DPO       | MIA-DPO-28k      | 60.3   | 53.7  | 57.0  |

---

> ### Author Response · Authors · 2025-08-07
> **Round 1**
>
> > Q2: The concern about the performance improvements stem mainly from data rather than optimization techniques.
>
> Thank you for this insightful question. We’d like to clarify our contributions by systematically addressing three key points: (1) the **meaningful performance gains** from our CcDPO method itself; (2) our MultiScope-42k preference **dataset is a core strength, not a weakness**; and (3) the **critical synergy** between the method and dataset that drives our results.
>
> ### **(1) CcDPO delivers meaningful gains as a standalone method**
> As shown in the table in our response to W2,Q2(1) and Q3, CcDPO method achieves a **maximum 3.7-point gain** on Mantis with MIA-DPO-28k and **maximum 4.2-point gain** on Mantis with MultiScope-42k. We consider this **improvement relatively significant**. Specific reasons include:
>
> - **Alignment with DPO literature standards**: A review of papers focused on improving DPO methods [1,2,3] indicates that **2 point improvements are regarded as notable**. For example, method [1] outperforms vanilla DPO by 2 points, while [3] achieves a maximum 1.3-point gain on general tasks—confirming that our method’s gains are meaningful in this context.
>
> - **Ablative validation of our two-stage design**: Comparing the last two rows in our W3,Q3 response table, swapping CcDPO’s optimization order (reversing the coarse-to-fine sequence) leads to a noticeable performance drop. In contrast, **the correct order yields a maximum 3.3-point improvement**, directly validating that our two-stage strategy drives these gains.
>
> - **Novel perspective contributions**: At the same time, we believe that the focus is not only on performance; our work itself offers new thoughts and perspectives. We **first analyzed multi-image understanding barriers** in current MLLMs (Fig. 2 of the main paper), identified **specific multi-image hallucinations**, and revealed inherent flaws in how these models handle multi-image tasks (Section 3)—insights we believe provide valuable guidance to the community. Building on this, we designed a coarse-to-fine two-stage DPO process that **mirrors human reasoning**: first grasping the global context, then zooming in on task-specific details. We’ve demonstrated its effectiveness and hope this framework informs future solutions for multi-image tasks.
>
>
> ### **(2) MultiScope-42k is a strength, not a limitation**
> High-quality preference data is widely recognized as a core contribution in DPO research, and MultiScope-42k advances this tradition:
>
> - **Precedent in DPO literature**: Many influential works prioritize preference data construction. For example:
>   - LLaVA-RLHF [4], HA-DPO [5], and POVID [6] use external models (e.g., GPT-4, GPT-4V) to generate or manipulate negative samples for preference datasets.
>   - MIA-DPO, a pioneering multi-image work, centers its contribution on DPO data construction (e.g., pre-filtering rejected responses, designing DPO questions) with no changes to DPO optimization.
>   - Overall, these studies are **valued for their data construction methodologies**, which have provided critical insights to the community. Following this precedent, we argue that high-quality multi-image preference datasets like **"MultiScope-42k" and our construction pipelines** constitute meaningful contributions to the field in their own right.
>
> - Furthermore, we would like to emphasize three distinct advantages of CcDPO’s dataset construction:
>   - **Effective proxy tasks**: Unlike MIA-DPO (which links questions to single images) tends to resolve hallucinations in image referencing, we use multi-image captioning as a proxy to enhance contextual understanding, specifically addressing the core multi-image hallucinations (context omission, conflation, detail misinterpretation) (Section 3, line 158).
>   - **Efficient preference pair generation**: Leveraging the identified hallucinations and benefiting from structured caption concatenation, we efficiently built two-stage preference data (global context + local details) at low cost. Negative samples—generated via caption swapping, truncation, or mismatched regions—precisely simulate critical multi-image hallucinations (Sections 3.1, 3.2).
>   - **Scalability**: Our construction easily extends to other domains/modalities (e.g., video, with a 0.7% improvement; see our response to reviewer wac8) and scales effectively with larger datasets (see our response to W2,Q2(3)).
>
> ---
> Reference
>
> [1] mdpo: Conditional preference optimization for multimodal large language models.
>
> [2] Symmetrical Visual Contrastive Optimization: Aligning Vision-Language Models with Minimal Contrastive Images
>
> [3] CHiP: CROSS-MODAL HIERARCHICAL DIRECT PREFERENCE OPTIMIZATION FOR MULTIMODAL LLMS
>
> [4] Aligning large multimodal models with factually augmented rlhf.
>
> [5] Enhancing lvlms through hallucination-aware direct preference optimization.
>
> [6] Aligning modalities in vision large language models via preference fine-tuning.

---

> ### Author Response · Authors · 2025-08-07
>
> ### **(3) Synergy between method and dataset is also critical**
> CcDPO’s value lies not in isolated components but in their integrated design:
>
> - The two-stage DPO framework requires accurate preference optimization for **both global contexts and local details**—needs directly supported by MultiScope-42k, which p**rovides context-details positive/negative samples** tailored to this structure. Together, they deliver a maximum 9.2-point improvement on Mantis. **Decoupling them reduces effectiveness**, explaining why CcDPO yields relatively **smaller gains on other datasets (which lack samples covering both global and local info)**.
>
> - As confirmed by our ablative analysis (the W3,Q3 table), **both the coarse-to-fine order of CcDPO and the combined use of its two stages are critical**: reversing this order or using only one stage diminishes performance, while the correct sequence delivers a 3.3-point gain. This underscores that neither the method nor the dataset alone is sufficient—their coordination is key.
>
> - Additionally, the ablation studies in **Tables 5 and 6 of the main paper provide detailed analyses** of CcDPO's individual stages and their synergistic effects. Specifically, standalone Context-Level DPO training already yields an overall **2.3-points** improvement, while incorporating Needle-Level DPO training further enhances performance by a maximum of **6.1 points**—highlighting how the **sequential integration of both stages amplifies gains beyond their individual contributions**.
>
> In summary, the value of CcDPO cannot be reduced to either a new DPO optimization technique or a new dataset **alone**. Instead, it arises from the **integrated design** of both components: the two-stage DPO training framework and the carefully
> constructed MultiScope-42k dataset. Their synergy is critical to the overall performance.

---

> > ### Comment · Reviewer_ut67 · 2025-08-07
> > **Thanks for quick response.**
> >
> > Thank you very much for your detailed and thoughtful response. I am interested in better understanding the contributions of CcDPO when the influence of the MultiScope-42k dataset is decoupled.
> >
> > As part of our effort to better understand and verify the foundation of your proposed improvements, may I kindly request that you explicitly include the performance of MIA-DPO trained on Qwen2-VL using MultiScope-42k, evaluated across the main benchmarks you have adopted (e.g., MuirBench, BLINK, Mantis, NLVR2, and Q-Bench2)?
> >
> > This would greatly help reviewers disentangle the contributions of your two-level DPO strategy from those of the base model and dataset.

---

> ### Author Response · Authors · 2025-08-08
> **Further Discussion with reviewer ut67 （Round 2）**
>
> ----------------------------
> We sincerely appreciate Reviewer ut67 for your valuable feedback on our two-level DPO strategy. Your insights on the ablation study have strengthened our work’s robustness.
> > Disentangling Contributions of the Two-Level DPO Method and Base Models
>
> Following your suggestion, we have conducted additional ablation experiments on Qwen2-VL (and extended them to Qwen2.5-VL for generalizability). As shown in the table below, it delivers robust average gains of **2.2 points on LLaVA-OV**, **1.3 points on Qwen2-VL**, and **1.4 points on Qwen2.5-VL**, collectively confirming the method’s applicability across base models.
>
> **Table 1: New Ablation Study on Qwen2-VL (Conducted per Reviewer Request)**
>
> | Method | Training Dataset | MuirBench | Mantis | BLINK | NLVR2 | Q-Bench2 | Average |
> |--------|------------------|-----------|--------|-------|-------|----------|---------|
> | **Qwen2-VL** | - | 40.5 | 65.9 | 53.4 | 84.8 | 74.5 | 63.8 |
> | + SFT | MultiScope-42k Data | 43.1 | 64.9 | 54.7 | 85.2 | 74.1 | 64.4 |
> | + MIA-DPO | MultiScope-42k Data | 43.5 | 68.2 | 55.6 | 85.5 | 74.9 | 65.5 |
> | + Needle-Level -> Context-Level | MultiScope-42k Data | **45.0** | 68.5 | 55.1 | 86.0 | 75.4 | 66.0 |
> | + Context-Level -> Needle-Level (Ours) | MultiScope-42k Data | 44.8 | **69.1** | **56.5** | **86.4** | **77.0** | **66.8** |
> | **Qwen2.5-VL** | - | 42.9 | 71.8 | 55.0 | 87.4 | 75.3 | 66.5 |
> | + MIA-DPO | MultiScope-42k Data | 43.8 | 71.4 | 57.5 | 87.8 | 74.7 | 67.0 |
> | + Context-Level -> Needle-Level (Ours) | MultiScope-42k Data | **45.0** | **73.3** | **58.8** | **88.1** | **76.6** | **68.4** |
>
> **Table 2: Original Ablation Study on LLaVA-OV (Response to Q3)**
>
> | Method                          | Dataset         | MuirBench | Mantis | BLINK | Q-Bench2 | Average |
> |---------------------------------|-----------------|-----------|--------|-------|----------|---------|
> | LLaVA-OV                        | -               | 42.5      | 60.4   | 51.1  | 73.8     | 56.9    |
> | SFT                             | MultiScope-42k  | 45.4      | 64.9   | 53.4  | 75.7     | 59.9    |
> | MIA-DPO | MultiScope-42k  | 46.2      | 65.4   | 55.1  | 76.1     | 60.7    |
> | Needle-Level -> Context-Level   | MultiScope-42k  | 46.5      | 66.3   | 54.1  | **77.8** | 61.2    |
> | Context-Level -> Needle-Level (Ours) | MultiScope-42k  | **48.6** | **69.6** | **55.9** | 77.6 | **62.9**    |
>
> These comprehensive results allow us to draw the following conclusions:
> - **Intrinsic Advantage and Cross-Model Generalizability:** All strategies (SFT, one-stage DPO, and CcDPO) use the same MultiScope-42k dataset, so performance differences directly reflect the impact of the DPO method.
>   - Compared to the one-stage vanilla DPO (MIA-DPO), our CcDPO strategy improves the average performance by **1.3 points on Qwen2-VL** (65.5 → 66.8) and **1.4 points on Qwen2.5-VL** (67.0 → 68.4). These gains are consistent across benchmarks, confirming that CcDPO’s advantages are intrinsic to its two-level design.
>   - CcDPO consistently improves performance across LLaVA-OV, Qwen2-VL, and Qwen2.5-VL—indicating **it works well across varying base capabilities of the underlying models**. These gains are meaningful given the Qwen series’ inherent strong long-context ability, underscoring the broad generalizability of our approach.
> - **The "Coarse-to-Fine" Order is Crucial**: We further tested reverse-order optimization (Needle-Level → Context-Level) on Qwen2-VL, which yielded a **lower average (66.0) than our CcDPO (66.8)**. Interestingly, we found that this reverse strategy performed better on MuirBench. We hypothesize this may be due to Qwen2-VL's inherent strength in high-resolution image understanding—in this specific benchmark, greater emphasis on contextual information enables it to accomplish complex tasks more effectively.
>
>
> We hope this additional validation helps address your concerns regarding our method’s contributions. We would be grateful if you would consider these new findings in your re-evaluation of our work. Please feel free to reach out if further clarification is needed—we are happy to respond promptly. Thank you again for your time and consideration.

---

> ### Comment · Reviewer_ut67 · 2025-08-08
>
> Thank you for the additional experimental results. I have the following questions about the results.
>
> 1. You attributed the underperformance of LLaVA-OV + MIA-DPO compared to SFT to the fact that SFT was trained on MultiScope-42k, whereas LLaVA-OV + MIA-DPO used only a 28k dataset. However, Table 1 shows that MIA-DPO underperforms both vanilla LLaVA-OV and Qwen2-VL on several benchmarks, including MuirBench, Mantis, NLVR2, and MIBench. In contrast, the original MIA-DPO paper reports significant improvements over vanilla LLaVA-v1.5 on benchmarks such as MMMU, BLINK, Mantis, NLVR2, and MVBench. These discrepancies raise concerns about whether the reproduction of MIA-DPO was successful and faithful to the original implementation.
>
> 2. In your first-round results, MIA-DPO showed a notable performance gain with LLaVA-OV after training on the MultiScope-42k dataset (from 41.4 to 46.5). However, in the second-round experiments with Qwen2-VL, MIA-DPO performs worse after training on the same dataset.
>
>
> Qwen2-VL
> | methods | datasets | MuirBench|Mantis | BLINK | NLVR2 | Q-Bench2 |
> |---------|----------|--------|--------|-------|-------|----------|
> |MIA-DPO| MIA-DPO-28k|40.1|69.1|54.5|84.5|75.6|
> |MIA-DPO|MultiScope-42k|43.5(+3.4)|**68.2(-0.9)**|	55.6(+1.1)|	85.5(+1.0)|	**74.9(-0.7)**|
> |CcDPO|MultiScope-42k|44.8|	69.1	|56.5|	86.4	|77.0
>
> LLaVA-OV
> | methods | datasets | MuirBench|Mantis | BLINK  | Q-Bench2 |
> |---------|----------|--------|--------|-------|----------|
> |MIA-DPO| MIA-DPO-28k|41.4|60.3|53.7|74.0|
> |MIA-DPO|MultiScope-42k|46.2(+4.8)|65.4(+5.1)|55.1(+1.4)|76.1(+2.1)|
> |CcDPO|MultiScope-42k|45.0|	68.5|	55.1|	75.4|
>
>  **Interestingly, the metrics where MIA-DPO underperforms after training on MultiScope-42k are those where it previously outperformed CcDPO—even when trained on the smaller 28k dataset.** Could you clarify the cause of this performance degradation?

---

> ### Author Response · Authors · 2025-08-08
> **Further Discussion with reviewer ut67 (Round 3)**
>
> -----------
> > Q1. On the Discrepancy with MIA-DPO’s Original Results
>
> We **recognize MIA-DPO as a significant and pioneering method in this area**, and for that reason, we believe it is important to clarify the context of our experimental results with the following three points:
> 1. First, the **base models used in our work are entirely different** from those in the original MIA-DPO paper. The original paper employed LLaVA-v1.5 and InternLM-XC2.5, whereas our research utilizes LLaVA-OV and Qwen2-VL. Given the significant differences in foundational capabilities and architectures across these base models, it is a normal and expected outcome that the performance gains from any given optimization method will vary.
> 2. Second, our reproduction process was conducted with **rigorous adherence to academic standards** to ensure the validity of our results. Specifically:
>    - We first attempted a complete replication of the LLaVA-v1.5 + MIA-DPO results using the official public code released by the MIA-DPO authors. However, we encountered technical challenges that **prevented a successful replication** within our specific experimental setup. **We have maintained full records of this reproduction attempt to ensure full transparency.**
>    - Consequently, to ensure a fair comparison against all other baselines presented in our paper, we standardized the experimental frameworks. **For LLaVA-OV, we used the official llava-next implementation**, and for **Qwen models, we used the LLaMA-Factory framework**. All variables were strictly controlled throughout every training process. The results reported in our manuscript are the outcome of these meticulously controlled experiments.
> 3. Third, to verify the accuracy of our MIA-DPO reproduction, the **complete training code is available in the supplementary materials.** We encourage the reviewer to validate our reported results. As detailed in the provided README file in the previous rebuttal, the training process can be initiated with the following command:
>     > cd LLaVA-NeXT
>
>     > bash scripts/train/dpo_ov7b.sh

---

> ### Author Response · Authors · 2025-08-08
> **Further Discussion with reviewer ut67 (Round 3)**
>
> -----------
> > Q2(1). On Inconsistent Performance Trends: Degradation in Qwen2-VL (Mantis and Q-Bench2) vs. Improvements in LLaVA-OV
>
> We provide the following analysis to explain the inconsistent performance of MIA-DPO on Qwen2-VL versus LLaVA-OV.
>
> 1. **Inherent differences in the base model.** It is well-documented in the literature [1,2,3,4,5] that **the same method often yields inconsistent results across different base models**. This variability is a common phenomenon in the field. This phenomenon is also present in the original paper of MIA-DPO As their public results show (see table, left panel), the exact same method (MIA-DPO) produced a strong **5.6-point gain** on InternLM-XC2.5 but a **1.1-point loss** on LLaVA-v1.5.
>
> |Base Model|Datasets|MVBench (Results in MIA-DPO paper)|Base Model|Datasets|Q-Bench2|
> |-|-|-|-|-|-|
> |LLaVA-v1.5| HA-DPO|40.6|Qwen2-VL| MIA-DPO|75.6|
> |LLaVA-v1.5| MIA-DPO|**39.5 (-1.1)**| Qwen2-VL|MultiScope-42k| **74.9 (-0.7)**|
> |InternLM-XC2.5| HA-DPO|58.0 | LLaVA-OV| MIA-DPO |74.0|
> |InternLM-XC2.5| MIA-DPO|**63.6 (+5.6)**| LLaVA-OV|MultiScope-42k|**76.1 (+2.1)**|
>
> 2. While such variability across base models is well-observed, we attempt to offer our hypothesis for these fluctuations: **base model capability mismatches with optimization setup.** Qwen2-VL, which outperforms LLaVA-OneVision significantly per OpenCompass’s leaderboard, has stronger inherent image understanding. This strength may backfire when paired with the **two-stage MultiScope-42k dataset** (emphasizing global and local details) via **one-stage MIA-DPO**, creating a **mismatch**. Qwen2-VL’s robust understanding could let it distinguish these demands, but the one-stage method may force it to balance competing objectives simultaneously. This can trigger internal conflicts, dilute the model’s focus, and reduce optimization efficiency.​
>
> 3. **Overall performance remains robust.** Against this backdrop of inherent variability, we consider the **overall performance** to be the more robust indicator of a method’s value. As shown in the following table, while the MIA-DPO baseline showed fluctuations on individual benchmarks, its **overall average score on Qwen2-VL still improved by 0.8 points (65.5 vs.64.7 ).** Building on this,  our CcDPO method not only achieves a more substantial overall **gain of 2.0 points (from 64.7 to 66.7)** but also delivers these **improvements more consistently** across all benchmarks. This underscores the effectiveness of our proposed CcDPO and the MultiScope-42k dataset.
>
> |Base Model|methods|datasets|MuirBench|Mantis|BLINK|NLVR2|Q-Bench2|Average|
> |-|-|-|-|-|-|-|-|-|
> |Qwen2-VL|MIA-DPO (One-Stage)|MIA-DPO-28k|40.1|**69.1**|54.5|84.5|**75.6**|**64.7**|
> |Qwen2-VL|MIA-DPO (One-Stage)|MultiScope-42k|43.5|**68.2 (-0.9)**|55.6|85.5|**74.9 (-0.7)**|**65.5 (+0.8)**|
> |Qwen2-VL|CcDPO (two-Stage)|MultiScope-42k|44.8|**69.1 (+0.0)**|56.5|86.4|**77.0 (+1.4)**|**66.7 (+2.0)**|
>
> -----------
> > Q2(2). Clarification on a Misinterpreted Comparison: “Interestingly, the metrics (Mantis and Q-Bench2) where MIA-DPO underperforms after training on MultiScope-42k are those where it previously outperformed CcDPO.”
>
> Regarding this comment, it seems to be due to a **misunderstanding of the result table.**
>
> - We note that the reviewer's initial comments **accidentally confused our results** (specifically, mistaking our ablation study baseline [needle level → context level] for the results of our CdDPO method in the Qwen2-VL table, with **77.0 misinterpreted as 75.4), which may have led to incorrect conclusions**. We appreciate the reviewer's **subsequent correction of this number in their comment.** With this corrected data (as shown in the table above), the claimed phenomenon—`the metrics where MIA-DPO underperforms... being those where it previously outperformed CcDPO`—**did not actually occur.**
>
> - We further emphasize that performance should not be the sole focus. Our work contributes novel insights by: (1) **conducting the systematic analysis of multi-image understanding barriers** in MLLMs (Figure 2 in the main paper), **identifying specific multi-image hallucinations**, and revealing inherent flaws in models’ handling of multi-image tasks (Section 3)—insights that offer valuable guidance to the community; (2) proposing a **coarse-to-fine two-stage DPO framework** that emulates human reasoning (global context comprehension followed by task-specific detail analysis). Its effectiveness has been demonstrated, and we anticipate this framework will inform future multi-image task solutions.
>
> [1] CLIP-DPO: Vision-Language Models as a Source of Preference for Fixing Hallucinations in LVLMs
>
> [2] Automated Multi-level Preference for MLLMs
>
> [3] Migician: Revealing the Magic of Free-Form Multi-Image Grounding in Multimodal Large Language Models
>
> [4] VideoPASTA : 7K Preference Pairs That Matter for Video-LLM Alignment
>
> [5] RLHF-V: Towards Trustworthy MLLMs via Behavior Alignment from Fine-grained Correctional Human Feedback

---

> ### Comment · Reviewer_ut67 · 2025-08-09
>
> The conclusions I can draw from the paper and the rebuttal would be as follows:
>
> ### (1) MultiScope-42k is a strong dataset that boosts performance across SFT, MIA-DPO, and CcDPO, as evidenced by the following results:
>
> LLaVA-OV
> | methods | datasets | MuirBench|Mantis | BLINK  | Q-Bench2 |
> |---------|----------|--------|--------|-------|----------|
> | SFT| MIA-DPO-28k|42.6|59.0|	52.3|	74.0|
> | SFT| MultiScope-42k|45.4(+2.8)	|64.9(+5.9)	|53.4(+1.1)|75.7(+1.7)|
> |MIA-DPO| MIA-DPO-28k|41.4|60.3|53.7|74.0|
> |MIA-DPO|MultiScope-42k|46.2(+4.8)|65.4(+5.1)|55.1(+1.4)|76.1(+2.1)|
> |CcDPO|MIA-DPO-28k|42.9|	62.7	|53.2|	74.8|
> |CcDPO|MultiScope-42k|48.6(+5.7)|	69.6+(6.9)|	55.9(+2.7)|	77.6(+2.8)|
>
> ### (2) MIA-DPO was not reproduced as faithfully as CcDPO, as evidenced by the following observations: MIA-DPO underperforms the vanilla model:
>
> LLaVA-OV
> | methods | datasets | MuirBench|Mantis | NVLR2| MIBench |
> |---------|----------|--------|--------|-------|----------|
> |Vanilla|MIA-DPO-28k|42.5|60.4|89.4|73.6|
> |MIA-DPO| MIA-DPO-28k|41.4(-1.1)|60.3(-0.1)|88.2(-1.2)|67.8(-5.8)|
>
> MIA-DPO **serves as the sole baseline in this paper, yet it performs even worse than the vanilla model**, contradicting the results reported in the original MIA-DPO paper.
>
>
> ### (3) CcDPO performs worse than MIA-DPO, even when trained on the superior MultiScope-42k dataset.
>
> Qwen2-VL
> | methods | datasets | MIRB|Mantis | Q-Bench2 |Average
> |---------|----------|--------|--------|-------|----------|
> |MIA-DPO| MIA-DPO-28k|61.4|69.1|75.6|64.5
> |CcDPO| MultiScope-42k|60.7(-0.7)|69.1|77.0(+1.4)|66.6(+2.1)|
>
> Given that MIA-DPO was not faithfully reproduced, **the actual contribution of the CcDPO algorithm appears marginal**, contrary to the authors’ claims.
>
> The main contribution of this paper lies in introducing a larger and more comprehensive dataset, MultiScope-42k. However, the authors consistently attempt to present CcDPO as the primary driver of performance gains—particularly in Figure 1, where the comparison does not control for the confounding factor of using different datasets. Their conclusions, along with the provided non-reproducible code, indicate that the actual improvements stem from MultiScope-42k. This work would be more appropriately submitted to the NeurIPS Datasets track rather than the Main track, which is precisely why NeurIPS distinguishes between these two tracks.

---

> ### Author Response · Authors · 2025-08-09
> **Further Discussion with reviewer ut67 (Round 4)**
>
> Thank you for raising these points. Let's go through them one by one
> > (1) MultiScope-42k is a strong dataset, and CcDPO algorithm appears marginal:
>
> - We acknowledge that the performance gain from the MultiScope-42k dataset is **relatively more pronounced** than the gain from the methodological ordering alone. However, as noted in our response to Q2, **recent DPO literature [1,2,3] considers improvements of 1–2 points to be significant. (It has been explained in question Q2 in round 1)**.
> - As confirmed by the ablation study below, our CcDPO design is **highly effective**. Our "Coarse-to-Fine" approach significantly outperforms the "Fine-to-Coarse" baseline and MIA-DPO, delivering performance gains of **1.9 and 2.3 points.**
>
> | Method    | Dataset         | MuirBench | Mantis | BLINK | Q-Bench2 | Average |
> |--------------|-----------------|-----------|--------|-------|----------|---------|
> | CcDPO | MIA-DPO-28k  | 42.9      | 62.7   | 53.2  | 74.8        | 58.4 (-4.6)   |
> | MIA-DPO | MultiScope-42k  | 46.2      | 65.4   | 55.1  | 76.1     | 60.7 **(-2.3)**    |
> | CcDPO (Fine-to-Coarse) | MultiScope-42k  | 46.5 | 66.3 | 54.1 | **77.8** | 61.1 **(-1.9)**    |
> | CcDPO (Coarse-to-Fine) (Ours) | MultiScope-42k  | **48.6** | **69.6** | **55.9** | 77.6 | 63.0 |
>
> - Furthermore, as mentioned in our Q2 response, influential papers [4,5,6] have made contributions by focusing on the construction of high-quality preference data without modifying the DPO algorithm itself, with works like **LLaVA-RLHF and MIA-DPO being accepted at main track**.  Therefore, a contribution should be evaluated **not only by the source of its performance gains** but also by **whether it analyzes a problem from a novel perspective and effectively solves it.**
>
> [1] mdpo: Conditional preference optimization for multimodal large language models.
>
> [2] Symmetrical Visual Contrastive Optimization: Aligning Vision-Language Models with Minimal Contrastive Images
>
> [3] CHiP: CROSS-MODAL HIERARCHICAL DIRECT PREFERENCE OPTIMIZATION FOR MULTIMODAL LLMS
>
> [4] Aligning large multimodal models with factually augmented rlhf.
>
> [5] Enhancing lvlms through hallucination-aware direct preference optimization.
>
> [6] Aligning modalities in vision large language models via preference fine-tuning.
>
> > (2) MIA-DPO was not reproduced as faithfully as CcDPO, as evidenced by the following observations: MIA-DPO underperforms the vanilla model:
>
> - The assertion that "MIA-DPO underperforms the vanilla model" is inaccurate. As shown in the table, our reproduction of MIA-DPO with the Qwen2-VL base model **outperforms the vanilla model on average (+0.6 points)**.
> - It is critical to note that the **base models used in our work are entirely different from those in the original MIA-DPO paper,**  which means the performance gains between the two studies are not directly comparable.
> - This discrepancy suggests that the MIA-DPO method may **be merely sensitive to the choice of the base model**;  **it cannot be used as direct evidence to claim our results are inauthentic.**
>
> | Models | datasets | MuirBench | MIRB  | BLINK  | Mantis       | NLVR2        | MIBench      | Q-Bench2     | Average      |
> |--|--|--|--|-|-|--|-|-|-|
> | Qwen2-VL   | - | 40.5  | 59.5 | 53.4  | 65.9   | 84.8         | 68.9         | 74.5         | 63.9         |
> | + MIA-DPO  | MIA-DPO-28k     | 40.1 (-0.4)  | 61.4 (+1.9)  | 54.5 (+1.1)  | 69.1 (+3.2)  | 84.5 (-0.3)  | 66.7 (-2.2)  | 75.6 (+1.1)  | 64.5 (+0.6)  |
>
> - We have **repeatedly provided the details of our reproduction efforts** and **shared the code to verify our results**. The reviewer's continued speculation about problems with our reproduction is unfounded.
>
> > (3) CcDPO achieves worse performance in comparison with MIA-DPO, even when CcDPO is trained with a better dataset MultiScope-42k.
>
> - This conclusion is incorrect. In fact, **CcDPO only underperforms MIA-DPO on a single benchmark** (MIRB with the Qwen2-VL base model).
> - Across all other benchmarks and both base models, **CcDPO demonstrates consistent and significant performance improvements** over MIA-DPO, resulting in substantial overall average gains of **+2.1 points (on Qwen2-VL) and +5.2 points (on LLaVA-OV)**, respectively.
>
> | Models   | datasets | MuirBench   | MIRB  | BLINK  | Mantis  | NLVR2 | MIBench  | Q-Bench2 | Average|
> |-|----|--|-|---|--|--|---|--|---|
> | Qwen2-VL + MIA-DPO   | MIA-DPO-28k     | 40.1        | 61.4        | 54.5        | 69.1        | 84.5        | 66.7        | 75.6        | 64.5        |
> | Qwen2-VL + CcDPO     | MultiScope-42k  | 44.8 (+4.7) | 60.7 (-0.7) | 56.5 (+2.0) | 69.1 (+0.0) | 86.4 (+1.9) | 71.9 (+5.2) | 77.0 (+1.4) | 66.6 (+2.1) |
> | LLaVA-OV + MIA-DPO       | MIA-DPO-28k   | 41.4        | 48.0        | 53.7        | 60.3        | 88.2        | 67.8        | 74.0        | 61.9        |
> | LLaVA-OV + CcDPO (Ours)       | MultiScope-42k  | 48.6 (+7.2) | 51.4 (+3.4) | 55.9 (+2.2) | 69.6 (+9.3) | 91.2 (+3.0) | 75.2 (+7.4) | 77.6 (+3.6) | 67.1 (+5.2) |

---

### Official Review · Reviewer_3cFy · 2025-07-03

**Clarity:** 3
**Significance:** 3
**Originality:** 3
**Rating:** 5
**Confidence:** 3

**Summary:**

This paper introduces a method for mitigating bias and hallucination in multi-image question answering (QA) with multimodal LLMs (MLLMs). The authors start by identifying the potential root causes of these errors: 1) context omission, 2) context conflation, and 3) misinterpretation. The authors constructed a preference optimization dataset to guide the MLLMs to avoid these issues. The authors also introduce the needle level preference optimization, which adds visual prompts to further guide the model towards preventing errors.

The proposed method is applied to open-sourced foundation LLMs and tested on multi-image QA benchmarks. Improvement in response quality is observed. The authors also provided ablation studies to justify the effectiveness of the proposed method over SFT and previous baselines.

**Questions:**

Using DPO to guide the model towards better spatial and sequential understanding seems to be an interesting direction. I am wondering in terms of spatial reasoning, whether the format of the added visual prompt has an impact on the quality. In the current text it is roughly described as "bounding boxes and key points".  Can the authors give more details on how the visual prompts are injected and what are the impact of different types of visual prompts?

**Ethical Concerns:**

["NO or VERY MINOR ethics concerns only"]

**Final Justification:**

The authors have addressed my remaining questions in the feedback. I am keeping my recommendation to accept.

**Limitations:**

Yes.

**Paper Formatting Concerns:**

N/A.

**Quality:**

3

**Strengths And Weaknesses:**

+ The problem of multi-image MLLMs is both important and interesting. It is great that the authors identify existing methods' problems with detailed analysis and empirical results. The analysis is technically sound to me.
+ Using multi-image captioning as a proxy to gauge the context understanding of multi-image MLLMs is an interesting take.
+ The proposed method directly addresses the issues identified with the existing methods. It uses light-weight DPO approaches so practitioners can easily adopt it.
+ Improvements in benchmark results are observed, and clear ablation studies are presented.

Weaknesses
- Details of how to construct visual prompts are missing.

---

> ### Author Rebuttal · Authors · 2025-07-31
>
> > **W1, Q1(1)**: More details on how to inject visual prompt.
>
> Apologies for any confusion caused. To clarify, the visual prompt construction process involves three key steps:
> - **Data Collection:** We start by gathering source data with point or bounding box annotations. Points are defined as [x, y], where x and y represent the coordinates of the point. Bounding boxes are defined as [x1, y1, x2, y2], where [x1, y1] and [x2, y2] correspond to the top-left and bottom-right corner coordinates, respectively.
> - **Prompt Injection:** Based on the collected text-based annotation coordinates, we convert them into visual prompts **(such as red points and or green rectangles for bounding boxes)** and overlay them onto the original image. This creates clear markings that guide the model's attention.
> - **Emphasizing Important Regions**: To further highlight the marked areas, we **add textual labels (e.g., “REF”) near the visual prompts**. Additionally, we incorporate corresponding textual prompts to emphasize the marked regions, such as: “Please provide a detailed description of the marked areas in each image. The marked areas are indicated by a green rectangle with a 'REF' label around them.
>
> More examples of visual prompts can be found in **Figures 1 and 7 of the supplementary material.**
>
> > **Q1(2):**: What are the effects of different types of visual cues on spatial reasoning?
>
> Thank you for this insightful question. To investigate how different types of visual prompts affect spatial reasoning, we conducted additional ablation studies using two additional visual prompt formats: **circle** and **segmentation mask**. The **circle** prompt is generated as an inscribed ellipse within the bounding box (bbox), while the **segmentation mask** is derived directly from annotations in the MDVP dataset. Below are the detailed results:
>
> | bbox | segmentation | point | circle |Mantis | BLINK | Q-Bench2 | Average |
> |:---:|:---:|:---:|:---:|:---:|:---:|:---:|:-----:|
> | | | | | 60.4 | 51.1 | 73.8 |61.8  |
> | ✔ | | ✔ | | 70.5 | 55.9 | 76.8 |67.7  |
> | ✔ | | | ✔ | 70.0 | 54.5 | **77.2** | 67.2  |
> | | ✔ | ✔ | | **71.0** | **56.8** | 76.3 | **68.0**  |
> | | ✔ | | ✔ | 69.1 | 55.1 | 76.0 | 66.7  |
> | | | ✔ | ✔ | 67.7 | 56.2 | 76.6 | 66.8  |
>
> From the results, it is evident that different visual prompts have a relatively minor impact overall, with performance variations mostly within 1 point. This highlights the robustness of our method, as it adapts well to various visual prompt formats. However, certain trends are noticeable across specific benchmarks:
>
> - **Fine-grained prompts** such as **segmentation masks** and **key points** are particularly beneficial for benchmarks focused on spatial reasoning, such as **Mantis** and **BLINK**. These prompts encourage models to focus on finer-grained areas, enabling more precise spatial descriptions.
>
> - **Coarse-grained prompts**, like **bounding boxes (bbox)** and **circles**, perform better in benchmarks that emphasize broader image-level comparisons, such as **Q-Bench2**. These prompts provide sufficient spatial guidance without overloading the model with excessive detail.
>
> Based on these findings, we chose the combination of **bounding boxes** and **key points** as the our final visual prompt format due to the following advantages:
> - **Balanced Spatial Representation**: This combination effectively captures global spatial relationships while incorporating local feature details, achieving the best overall spatial reasoning performance across benchmarks.
> - **Simplicity and Efficiency**: Bounding boxes and key points annotations are more simple to obtain, making data preparation significantly more cost-effective compared to segmentation masks, which may not always be available.

---

> > ### Comment · Reviewer_3cFy · 2025-08-05
> >
> > Thanks for the detailed response. It would be great to include the analysis of the visual prompt styles in the manuscript as guidance to practitioners of this method.

---

> > > ### Author Response · Authors · 2025-08-06
> > > **Thanks!**
> > >
> > > Thank you very much for your positive feedback and valuable suggestions! We completely agree with and have adopted your insightful suggestion to incorporate the analysis of visual prompt styles into our manuscript, which will greatly increase the robustness of our work. Additionally, your writing suggestions, such as further clarification on "how to inject visual prompts," have been incredibly helpful and significantly improved the readability of our paper. Thank you once again for your time and valuable contributions!

---

### Author Response · Authors · 2025-08-09
**General Response**

We sincerely thank all reviewers for your detailed and valuable comments. All reviewers **(3cFy, ut67, wac8, iyTW)** acknowledged the effectiveness of CcDPO, noting its consistent performance improvements across multi-image benchmarks such as MuirBench, BLINK, and Mantis. Our thorough analysis of multi-image hallucinations in existing MLLMs was praised for providing a solid foundation and focused direction for the methodology **(3cFy, ut67, wac8, iyTW)**. The proposed hierarchical DPO framework was highlighted for being methodologically sound and conceptually intuitive **(ut67, wac8)**. The clarity of the paper, the thoroughness of the experiments, and the detailed ablation studies were also commended **(3cFy, wac8, iyTW)**. Finally, the reviewers agreed that our work addresses an important and critical challenge **(3cFy, ut67)**.

> Experimental Additions

Based on the reviewers' feedback, we conducted several new experiments to provide more robust evidence:

- **[Reviewer ut67, iyTW]** We have provided detailed ablation studies demonstrating both the individual contributions and synergistic effects of the Context-Level and Needle-Level optimizations, showing they are complementary.

- **[Reviewer ut67]** We disentangled the contributions of our CcDPO method and our new MultiScope-42k dataset, showing through extensive ablations that CcDPO itself provides significant and consistent gains over baselines across multiple base models (LLaVA-OV, Qwen2-VL).

- **[Reviewer 3cFy]** We conducted ablation studies on various visual prompt styles (e.g., circle, segmentation mask) to validate the robustness of our method and provided clear details on their construction.

- **[Reviewer wac8]** We have extended CcDPO to video-related tasks, demonstrating its generalizability for sequence comprehension with minimal adaptation.

> Clarifications and Explanations

We also offered several key clarifications to address misunderstandings and provide further details:

- **[Reviewer wac8, iyTW]** We have clarified the novelty of CcDPO, detailing its key differences from prior work like MIA-DPO in terms of its full-context prompt design and cost-effective preference pair generation.

- **[Reviewer ut67]** We have analyzed and explained performance variations across different base models, providing hypotheses for the observed results and confirming the overall robustness of our approach.

- **[Reviewer iyTW]** We have added clear definitions for technical shorthand terms used in the paper to improve readability.

- **[Reviewer ut6T]** We have provided detailed reproducibility instructions in the rebuttal and committed to releasing fully documented code upon acceptance.

---

### Decision · Program_Chairs · 2025-09-17

**Decision:**

Accept (poster)

**Comment:**

This paper introduces Context-to-Cue Direct Preference Optimization (CcDPO), a hierarchical framework designed to reduce hallucinations in multi-image question answering. The method first operates at a "context level," using perturbation-based training to improve the model's overall understanding of the image sequence, preventing errors like context omission and conflation. It then applies a "needle level" optimization, which uses region-specific visual prompts to enhance the perception of fine-grained details and prevent misinterpretation. Trained on a newly constructed preference dataset, MultiScope-42k, the CcDPO method demonstrates consistent performance gains and reduces hallucinations across several multi-image benchmarks.

The paper received mixed reviews, with ratings of 2 and 5s. While reviewers generally acknowledged the novelty of the proposed CcDPO, the primary focus was on the reproducibility and significance of CuDPO. The authors actively engaged in the rebuttal phase, providing explanations for the provided source code and presenting additional numerical evidence to demonstrate the contribution of the CuDPO algorithm and the newly constructed MultiScope-42k dataset. Although reviewers may still have some reservations regarding the conclusions drawn from the data, the overall feedback remains positive. We recommend that the authors consider making further clarifications and incorporating the numerical data and discussions into the next version of the paper.